# Essential role of lattice oxygen in hydrogen sensing reaction

Jiayu Li[1], Wenzhe Si[2], Lei Shi[1], Ruiqin Gao [3] ✉, Qiuju Li [4] ✉, Wei An[1], Zicheng Zhao[1], Lu Zhang[1], Ni Bai[5], Xiaoxin Zou[1] & Guo-Dong Li [1] ✉

Understanding the sensing mechanism of metal oxide semiconductors is imperative to the development of high-performance sensors. The traditional sensing mechanism only recognizes the effect of surface chemisorbed oxygen from the air but ignores surface lattice oxygen. Herein, using in-situ characterizations, we provide direct experimental evidence that the surface chemisorbed oxygen participated in the sensing process can come from lattice oxygen of the oxides. Further density functional theory (DFT) calculations prove that the *p*-band center of O serves as a state of art for regulating the participation of lattice oxygen in gas-sensing reactions. Based on our experimental data and theoretical calculations, we discuss mechanisms that are fundamentally different from the conventional mechanism and show that the easily participation of lattice oxygen is helpful for the high response value of the materials.

Hydrogen, a high-energy-density energy carrier which can be produced from renewable sources (solar, hydro, wind), has recently received considerable interest in various fields, including new energy vehicles[1,2]. However, the easy escape and wide range of explosive concentration (4% to 75% when mixed with air) of hydrogen highly impedes the commercialization of hydrogen vehicles[3-5]. Developing high-performance on-board hydrogen sensors are necessary for the safe operation of hydrogen vehicles. Among various sensors, gas sensors made by metal oxide semiconductors have gained significant attention from researchers because of their low cost, small size, good stability, and easy production[6-8]. While, due to the complexity of the gas sensing process, the sensing mechanism of them is yet not totally understood, restraining the rational design of semiconductor sensing materials[9,10].

Currently, a widespread belief is that oxygen species ($O_2$, $O_2^-$, and $O^-$) plays a crucial role in the gas-sensing reaction, which is largely drawn from studies on the surface of metal oxide semiconductors[11-13]. The commonly accepted mechanism for hydrogen sensing is the chemisorbed oxygen mechanism, in which the hydrogen gas undergoes a redox reaction with the chemisorbed oxygen on the surface of the material, and the electrons transfer between the oxygen chemisorbed on the surface and conduction/valence band[14,15]. With this mechanism, all the oxygen species participating in the sensing process come from the air. Hence, a high sensing response can be obtained by increasing the content of chemisorbed oxygen, and this is supported by experimental findings that the sensing response of $In_2O_3$, $SnO_2$, ZnO, and so on[6-8,16,17]. However, this conventional mechanism has been challenged by a few observations in recent years. First, under vacuum conditions where in the absence of surface-adsorbed oxygen, the response of semiconductor sensors remains unchanged or even increases[18,19]. Second, there are instances where the gas-sensing performance is poor despite a high proportion of surface-adsorbed oxygen[20,21].

Herein, we provide experimental evidence for the involvement of lattice oxygen in the $H_2$ sensing mechanism. Using germanium-doped tin dioxide as a case, we utilize in-situ diffuse reflectance infrared Fourier transform spectroscopy (DRIFTS) and in-situ Raman spectroscopy to investigate the alteration of the surface during hydrogen gas-

[1]State Key Laboratory of Inorganic Synthesis and Preparative Chemistry, College of Chemistry, Jilin University, Changchun 130012, P. R. China. [2]State Key Joint Laboratory of Environment Simulation and Pollution Control, School of Environment, Tsinghua University, Beijing 100084, P. R. China. [3]School of Biological and Chemical Engineering, NingboTech University, No.1 South Qianhu Road, Ningbo 315100, P. R. China. [4]Department of Chemistry, College of Basic Medicine, Third Military Medical University (Army Medical University), Chongqing 400038, P. R. China. [5]School of Metallurgy Engineering, Jiangsu University of Science and Technology, Zhangjiagang 215600, P. R. China. ✉e-mail: gaorq@nbt.edu.cn; liqiuju93@tmmu.edu.cn; lgd@jlu.edu.cn

sensing process, finding the surface lattice oxygen participated in gas-sensing reactions. DFT calculations indicate that the elevation of the $p$-band center of oxygen plays a crucial role in enabling the participation of surface lattice oxygen in hydrogen gas-sensing reactions. 20%Ge-doped $SnO_2$ nanofibers show the highest response (S = 39.2 for 500 ppm $H_2$), well selectivity, and fast response speed (< 2 s for 0.1% $H_2$). Our work holds significant importance in furthering the understanding of the hydrogen gas-sensing mechanism and the development of gas-sensing materials.

## Results and Discussion
### Composition and structure analysis

The germanium-doped $SnO_2$ was synthesized by the electrospinning method using $SnCl_4 \cdot 5H_2O$ and bis(2-carboxyethyl germanium(IV) sesquioxide) (Ge-132) as precursors. In this work, as it has demonstrated the highest hydrogen sensitivity among the as-synthesized samples, therefore, most of the characterizations and discussions are focused on $Sn_{0.8}Ge_{0.2}O_2$ (named as SGO in the following part, $SnO_2$ is named as SO). The crystal structure of SO and SGO was first investigated by powder X-ray diffraction (XRD). As shown in Fig. 1a (Source data are provided as a Source Data file, the same below), the diffraction peaks of SGO are almost the same as those of rutile-SO besides a slight shift (e.g. 0.27°, 2-theta for the diffraction of (110)) of the diffraction peaks of SGO to higher angles, suggesting a small lattice shrinkage. This is caused by the incorporation of germanium with smaller ionic radius in SO ($Ge^{4+}$, 53 pm vs. $Sn^{4+}$, 69 pm). It is further proved by the crystal information of SGO and SO, which were obtained through Rietveld refinement and calibrated lattice shift (Supplementary Figs. 1 and 2, Supplementary Tables 1 and 2). In addition, compared to SO, the XRD diffraction peaks of SGO broadened significantly, which can be attributed to the reduced grain size of SO caused by the additional Ge species.

Subsequent TEM images have further proved this point (Supplementary Figs. 3 and 4).

To further reveal the regulation effect of additional Ge species on Sn-O bond, Raman spectra of both SO and SGO are recorded, as shown in Fig. 1b. SGO shows similar Raman bands with that of SO due to their similar rutile structure, which is agreement with XRD results. The Raman band at 632 $cm^{-1}$ can be assigned to the $A_{1g}$ symmetric stretching mode of Sn-O bonds, and the band central at 773 $cm^{-1}$ and 745 $cm^{-1}$ originates from asymmetric stretching mode ($B_{2g}$) of SO and SGO, respectively[22,23]. The Raman band central at around 550 $cm^{-1}$ is ascribed to the sub-bridging oxygen vacancies[24]. The comparison of the Raman spectra of SO and SGO can reveal that (i) the Raman bands for SGO, which are associated with Sn-O symmetric stretching mode, shift to lower frequency in comparison with that for SO, indicating the relatively weak bond strength and long bond length of Sn-O bond in SGO; (ii) compared with that of SO, the SGO exhibits more broadened Raman bands, indicating the increased structural distortion in SGO; (iii) SGO exhibits relatively stronger Raman bands at 550 $cm^{-1}$ than 632 $cm^{-1}$, indicating relative oxygen vacancy concentration ($A_{550}/A_{632}$) in SGO is higher than that in SO. Moreover, EPR spectra also confirm that SGO has more oxygen vacancies compared to SO (Supplementary Fig. 5). All these phenomena should originate from the electronic interaction between $Ge^{4+}$ and SO. DFT calculations also demonstrate that the introduction of Ge induces distortion in the SGO lattice, thereby increasing the likelihood of defect formation.

The elemental composition and valence of SO and SGO are studied by X-ray photoelectron spectroscopy (XPS). To ensure the reliability of XPS data, we performed data correction using carbon $1s$ peaks (Supplementary Fig. 6). The Ge $2p$ XPS spectrum in Supplementary Fig. 7 demonstrates that the oxidation state of Ge for SGO is $+4$[25]. The high-resolution Sn $3d$ XPS spectra of SO in Fig. 1c shows the binding energy of Sn $3d_{5/2}$ and Sn $3d_{3/2}$ at 486.6 eV and 495.0 eV, suggesting the

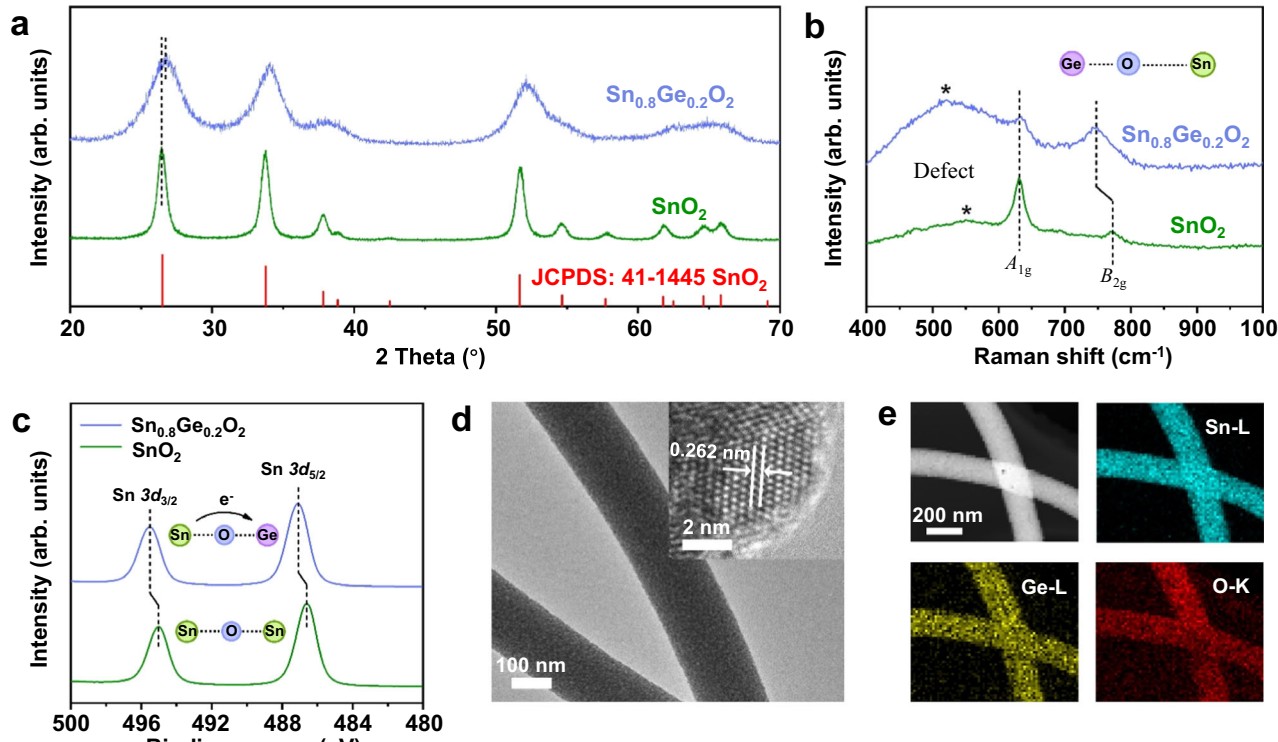

**Fig. 1 | structural characterizations of SGO and SO. a** XRD patterns of SGO and SO. **b** Raman spectra of SGO and SO. **c** Sn $3d$ XPS spectra of SGO and SO. Compared with SO, the Sn $3d$ peaks of SGO exhibit a slight shift toward higher binding energy, which may result from both the downshift of Fermi level and the increase in valence of Sn species in SGO. **d** TEM image and HRTEM image (inset) of SGO. **e** elemental mapping images of SGO form TEM. In all figures, SGO denotes $Sn_{0.8}Ge_{0.2}O_2$, and SO denotes $SnO_2$.

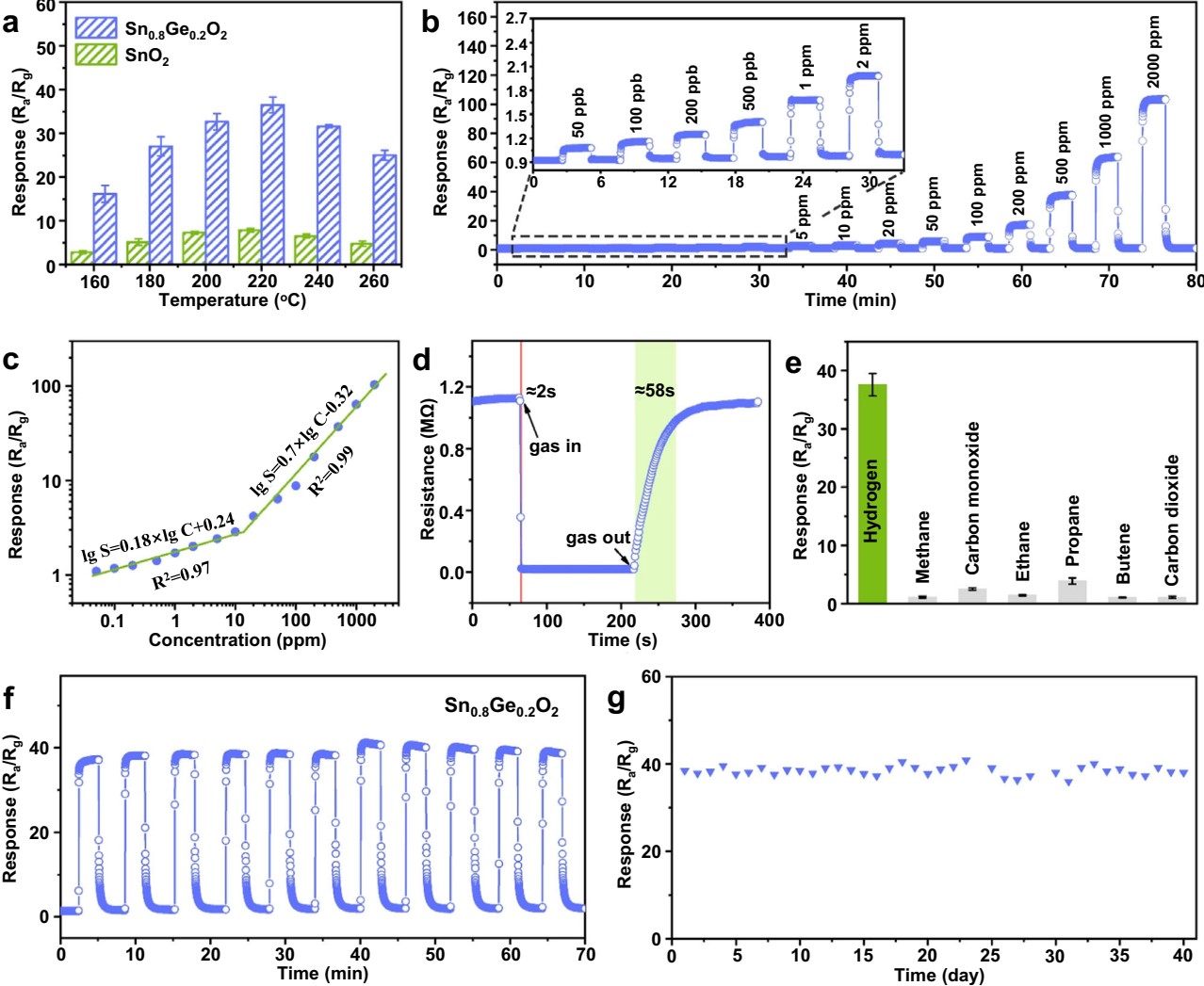

**Fig. 2 | Gas sensing properties of SGO. a** Responses of sensors based on SGO and SO to 500 ppm $H_2$ at different operating temperatures. **b** Dynamic response-recovery curves and **c** linear correlation of SGO-based sensor to 0.05-2000 ppm $H_2$. **d** response-recovery curve of SGO-based sensor to 1000 ppm $H_2$, **e** Selectivity to $H_2$ in the presence of $CH_4$, CO, ethane, propane, butane and $CO_2$. The gray part is the interference gas selected in this experiment. **f** dynamic cycling performance and **g** long-term stability of SGO-based sensor at 220°C. In all figures, SGO denotes $Sn_{0.8}Ge_{0.2}O_2$ and SO denotes $SnO_2$. The error bar represents standard deviations based on five measurements.

Sn species are $Sn^{4+}$ ions in SO[26-28]. Compared with SO, the Sn *3d* peaks of SGO exhibit a slight shift toward higher binding energy, which may result from both the downshift of Fermi level and the increase in valence of Sn species in SGO[11]. During gas sensing process, the Sn species with higher valence can catalyze the oxidation of $H_2$, thus improving the response speed of SGO[29,30].

The morphology and elemental distribution of SGO are characterized by scanning electron microscope (SEM) (Supplementary Fig. 8), transmission electron microscopy (TEM) (Fig. 1d and Supplementary Fig. 3) and elemental mapping (Fig. 1e). It can be observed that SGO consists of several micrometers long nanofibers assembled by nanoparticles with particle sizes in the range of 5-7 nm, which is much smaller than the nanoparticles in SO (15 nm, Supplementary Fig. 4). The smaller size of SGO particles may be attributed to the lattice contraction/mismatch caused by the introduction of germanium (0.262 nm vs. 0.264 nm, inset of Fig. 1d, Supplementary Fig. 9)[20,31]. Elemental mapping images of SGO (Fig. 1e) prove that the homogeneous distribution of all elements, including Sn, Ge and O, in the material. Energy dispersive X-ray spectroscopy (EDS) indicates the atomic ratio of Ge: Sn in SGO is 23%: 77%, which closely matches the feeding ratio (Supplementary

Fig. 10). Additionally, Supplementary Fig. 11-12 display $N_2$ adsorption−desorption isotherms of SO and SGO. The typical type IV isotherm indicates the presence of porous architectures. The BET surface areas of SGO were calculated to be 113 m²/g, three times higher than that of pure SO nanofibers (27 m²/g). The large surface area can provide more surface adsorption sites and more efficient gas diffusion, which is helpful to enhance response.

## Gas sensing properties

To evaluate $H_2$ sensing performance, we constructed miniaturized side-heated semiconductor gas sensors based on the obtained SGO and SO. The response values of the sensor vary depending on its working temperatures because the gas sensing reaction is strongly influenced by temperature[32,33]. To test sensors based on the SGO, and SO, they were exposed to 500 ppm $H_2$ at temperatures ranging from 160 to 260°C (Supplementary Fig. 13). The highest sensing response value for $H_2$ molecules was achieved at 220°C (Fig. 2a). Additionally, the sensing response initially increases and then decreases with the rise in temperature. With the increase in the operating temperature, the hydrogen molecules gain enough energy to overcome the activation energy barrier to react with the oxygen species. However, when

the temperature is too high, gas molecules move faster and desorb before they have time to participate in the gas-sensing reaction. Hence the responses of SGO exhibit a trend of "increase-maximum-decrease" as the operating temperature increases. The optimum working temperature for sensors based on the SGO was identified as 220 °C and this temperature was used in all subsequent tests.

To understand its sensing characters, the sensing response of the SGO to $H_2$ with concentrations ranging from 0.05 to 2000 ppm was measured at 220°C, as shown in Fig. 2b. The response values to $H_2$ are determined using the formula $S = R_a/R_g$, the response values to 0.05, 0.1, 0.2, 0.5, 1, 2, 5, 10, 20, 50, 100, 200, 500, 1000 and 2000 ppm $H_2$ are calculated to be 1.09, 1.17, 1.26, 1.41, 1.7, 2, 2.41, 2.86, 4.17, 6.4, 8.8, 17.87, 39.13, 63.93 and 103.73, respectively. Additionally, Fig. 2c shows the linear relationship between the logarithm of responses and the logarithm of $H_2$ concentrations of SGO-based sensor. The sensor responses show a gradual increase in rate ($\beta_1 = 0.18$) at low $H_2$ concentrations ranging from 50 ppb to 10 ppm. However, the rate of increase ($\beta_2 = 0.70$) becomes faster within the concentration range of 10 – 2000 ppm. Moreover, as shown in Fig. 2d, the SGO-based sensor has response time and recovery time of <2 s and 58 s, respectively, for 1000 ppm $H_2$. This meets the requirements set by the Society of Automotive Engineers (SAE). The concentration-dependent response time and recovery time of the SGO-based sensor are illustrated in Supplementary Fig. 14. The sensor exhibits a short response time (< 10 s) and recovery time (1 – 77 s) when exposed to $H_2$ concentrations ranging from 0.05 ppm to 2000 ppm at a temperature of 220°C. According to previous reports (Supplementary Table 3), the response time of SGO is significantly shorter compared to most hydrogen sensing materials, including those decorated with precious metals. In addition, the decrease and subsequently remain stable in resistance of SGO caused by hydrogen can be attributed to the reaction between SGO and hydrogen, which reaches its limit at the operating temperature and achieves an equilibrium state (see Supplementary Fig. 15 for a detailed discussion).

The selectivity of a hydrogen sensor is a crucial factor in determining its practical application. In this study, six kinds of possible interfering gases, including $CH_4$, CO, ethane, propane, butane and $CO_2$ are further tested. As shown in Fig. 2e, its response towards the interfering gases are all below 5, denoting the inadequate sensitivity of SGO towards them. Specifically, the sensor exhibits a response value of S = 39.2 for $H_2$, which is at least seven times greater than any of the interfering gases. This remarkable selectivity can be attributed to the enhanced adsorption capacity and greater reducibility of $H_2$ molecules on the surface of SGO.

The stability of SGO-based sensor was also carefully investigated. As recorded in Fig. 2f, the response and resistance of the sensor are almost constant value after 10 cycles of dynamic response tests, which indicates its well repeatability. Under the same test conditions, a long-term stability measurement was carried out for 40 days (Fig. 2g). The response of the sensor based on SGO within 40 days remains at a high level, suggesting prominent stability. The high repeatability and long-term stability of SGO-based sensors provide support for real-world applications.

The structure and morphology after the gas-sensing reaction was investigated to understand the stability of gas-sensing materials. The XRD patterns (Supplementary Fig. 16a) before and after the reaction showed almost no change, indicating the good stability of this material. Furthermore, the SEM image (Supplementary Fig. 16b) of SGO after the reaction also shows no significant changes compared to pristine sample. The HRTEM image (Supplementary Fig. 16c) of SGO after the gas-sensing reaction reveals a clear lattice structure, indicating there is no surface amorphization. In addition, as depicted in Supplementary Fig. 17, the signal from the oxygen vacancy remains relatively unchanged. This observation further supports the notion of the stability of SGO before and after gas-sensing reactions. Therefore,

it can be concluded that SGO maintains its crystal structure during the gas-sensing reaction.

We investigated the hydrogen-sensing property of various Ge-doped SO materials with different Ge: Sn molar ratios. The structural and morphological characterizations of these materials can be found in Supplementary Figs. 18 and 19 and Supplementary Table 4. We observed that the response value for $H_2$ increases as the Ge: Sn molar ratio increases, reaching a maximum at 2:8 (Supplementary Fig. 20). This non-linear correlation between the response value and the germanium content suggests that the remarkable response value of Ge-doped SO materials cannot be solely attributed to Ge doping reducing the particle size or increasing surface area. In other words, there may be an electronic structure-regulating effect in the Ge-doped SO system.

## Sensing mechanism

The conventional hydrogen sensing mechanism is chemisorbed oxygen mechanism, as illustrated in Supplementary Fig. 21[8,34]. Initially, oxygen is adsorbed directly on the surface of the gas-sensing material and captures electrons from the material to form chemisorbed oxygen. Upon hydrogen introduction, the hydrogen reacts directly with the chemisorbed oxygen, resulting in the formation of water molecules that gradually detach from the material's surface. The electrons captured by chemisorbed oxygen are then released back into the gas-sensing material. The amount of oxygen adsorbed on the surface plays a crucial role in determining the response value of the gas-sensing material. When more oxygen is adsorbed on the surface and participates in the gas-sensing reaction, the resistance of the gas-sensing material will change greatly, resulting in a higher response value.

Herein, we investigated the effect of energy level structure on chemisorbed oxygen. Reflected by work functions, Fermi levels of SGO and SO are identified as −5.2 and −5.6 eV. As revealed by VB XPS, spectrum of SGO presents a shift of about 0.4 eV to higher band energy compared to that of SO (Supplementary Fig. 22). The result, coupled with previous literature which shows VB edge position of SO is -8.0 eV, reveals that VB edge position of SGO is −7.6 eV[35]. Because band gap energy of SGO and SO are both 3.6 eV (Supplementary Fig. 23), the conduction band (CB) edge position of SGO and SO are determined to be −4 eV and −4.4 eV (Fig. 3a), respectively. This result is consistent with previous reports[36].

In the case of $n$-type semiconductors like SO and SGO, the primary carriers involved in gas sensing are electrons in the conduction band[37,38]. Higher conduction band bottom indicates that the electrons in the conduction band have higher energy with a larger energy level difference between oxygen molecules and semiconductors (Fig. 3b). Thus, oxygen molecules can capture more free electrons from the conduction band of SGO and form more adsorbed oxygen (Fig. 3c). When hydrogen is used in gas sensing with SGO, it reacts with more chemisorbed oxygen and releases more electrons to the conduction band of SGO compare with SO, resulting in an increased response value (Fig. 3d).

In order to explore the correlation between the elevation of the bottom of the conduction band and the augmentation of chemisorbed oxygen, experiments involving O $1s$ XPS and $O_2$-TPD were conducted on SGO and SO. The O $1s$ XPS spectra were analyzed and split into three distinct peaks, namely lattice oxygen ($O_L$), hydroxyl oxygen ($O_H$), and absorbed oxygen species ($O_C$)[39,40]. According to the displayed results, the percentages of absorbed oxygen species are 25% and 16% for SGO and SO, respectively (Fig. 3e and Supplementary Fig. 24). As shown in $O_2$-TPD profiles of SGO and SO (Fig. 3f), the green line representing SO shows a weak $O^-$ (ad) desorption peak between 300-400°C. On the other hand, the blue line representing SGO exhibits a very strong desorption peak. This indicates that there are considerably more chemisorbed monatomic species $O^-$ on the surface of SGO compared to SO. Moreover, we standardized the amount of adsorbed oxygen

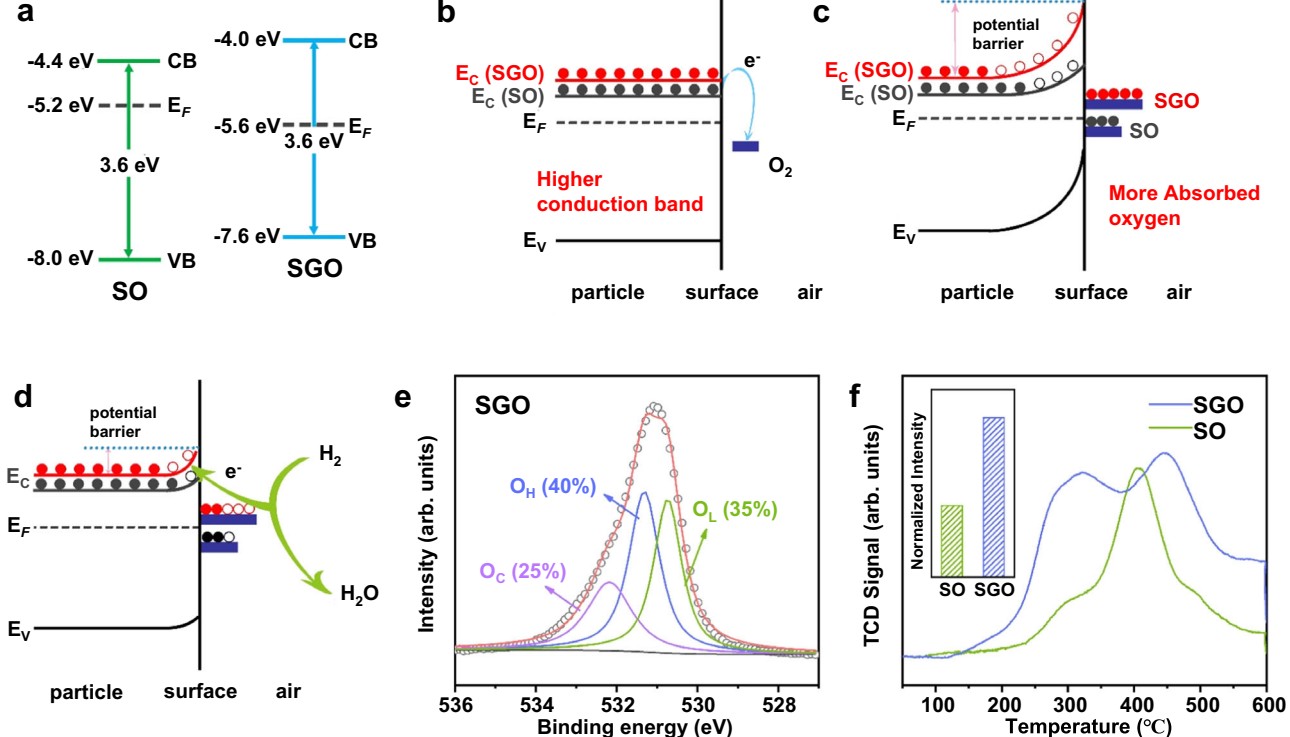

**Fig. 3 | Electronic structure of SGO. a** Schematic illustration of the energy band diagram of SO and SGO. **b**–**d** Schematic gas sensing mechanism of SO and SGO: in vacuum (**b**), in air (**c**), and exposure to $H_2$ (**d**). CB, VB, and $E_F$ denote minimum conduction band energy, maximum valence band energy, and Fermi level energy, respectively. The black and red dots represent free electrons in the conduction band of SO and SGO, respectively. **a**–**d** show that a Higher conduction band bottom will increase adsorbed oxygen and further increase response value. **e** O *1s* XPS spectrum of SGO. **f** $O_2$-TPD profiles of SO and SGO, with the normalized amount of $O_2$ adsorbed onto the surface of SO and SGO (inset), normalized intensity represents $O_2$-TPD signal area under 50–600°C. In all figures, SGO denotes $Sn_{0.8}Ge_{0.2}O_2$, SO denotes $SnO_2$, CB denotes the conduction band, and VB denotes the valence band.

measured by $O_2$-TPD and found that SGO adsorbed approximately twice as much oxygen as SO, which basically matches the XPS results. However, the increase in SGO response value cannot be solely attributed to the increase in adsorbed oxygen, as the percentage increase in adsorbed oxygen is significantly lower than the percentage increase in the response value.

To eliminate the influence of surface-adsorbed oxygen, we conducted additional research on the response of SGO to hydrogen in an Ar environment. In general, when surface adsorbed oxygen is a significant factor affecting the performance of gas-sensing materials, moving it to an Ar environment causes a significant decrease in the response of the gas-sensing material to the target gas, or even no response at all[25,41]. Unexpectedly, SGO shows a response value that remains the same or even higher under Ar conditions compared to that in air (Supplementary Fig. 25)[18,19]. To further verify this phenomenon, we have investigated various gases, including CO, formaldehyde, ethanol, acetone, and others, which are able to react with chemisorbed oxygen. The results indicate that the response values of most gases in the argon environment are higher compared to those in the air environment (Supplementary Fig. 26).

In order to address these questions, we conducted a study on the dynamic evolution process of the SGO surface during the hydrogen-sensing reaction. We first investigated the surface state of gas-sensing materials at varied temperatures by in-situ Raman. As shown in Fig. 4a and b, the peaks corresponding to lattice vibration ($A_{1g}$ and $B_{2g}$) weaken gradually, while the peaks related to surface oxygen vacancies enhance gradually with the increasing temperature[42]. Hence, the surface lattice oxygen certainly converts into other oxygen species under increased temperature.

To verify the results obtained by in-situ Raman, in-situ DRIFTS were further carried out to investigate the changes in surface-adsorbed oxygen at different temperatures (Supplementary Fig. 27). As illustrated in Fig. 4c, no peaks appeared in the in-situ DRIFTS at 50 °C, while upon heating, two peaks emerged at 803 and 1055 cm$^{-1}$. They originated from two kinds of absorbed oxygen, i.e. O$^-$ (803 cm$^{-1}$) and O$_2^-$ (1055 cm$^{-1}$)[43,44]. It is noteworthy that the peak represented O$^-$ (803 cm$^{-1}$) exhibits a pronounced enhancement with the elevation of the heating temperature, demonstrating that more absorbed oxygen generates on the surface of SGO with the increasing operating temperature[45,46]. Combined with recent reports involving lattice oxygen in catalytic reactions and the experimental results of in-situ Raman, we deduce that this phenomenon could be attributed to the conversion of surface-latticed oxygen to adsorbed oxygen, along with generating surface oxygen vacancies. In addition, the normalization of peak intensities reveals that as the operating temperature increases, the O$_2^-$ content decreases while the O$^-$ content increases gradually (Fig. 4d). At an operating temperature of 220°C, the primary oxygen species adsorbed on the surface should be O$^-$, as reported previously[47,48].

To reveal the interaction between hydrogen and the SGO, in-situ DRIFTS was further carried out in $H_2$/Ar at different working temperatures. As shown in Fig. 4e, the DRIFTS curves obtained in air are identified as the original background. Interestingly, with the introduction of $H_2$, two ravines represented lost chemisorbed oxygen displayed at 803 and 1055 cm$^{-1}$, while the ravines gradually are filled up after being exposed to air again. The oxygen adsorbed on the surface of the gas-sensing material is consumed upon hydrogen is introduced into the system. However, oxygen species will be re-adsorbed on the material's surface after the sensor is exposed to air again[49].

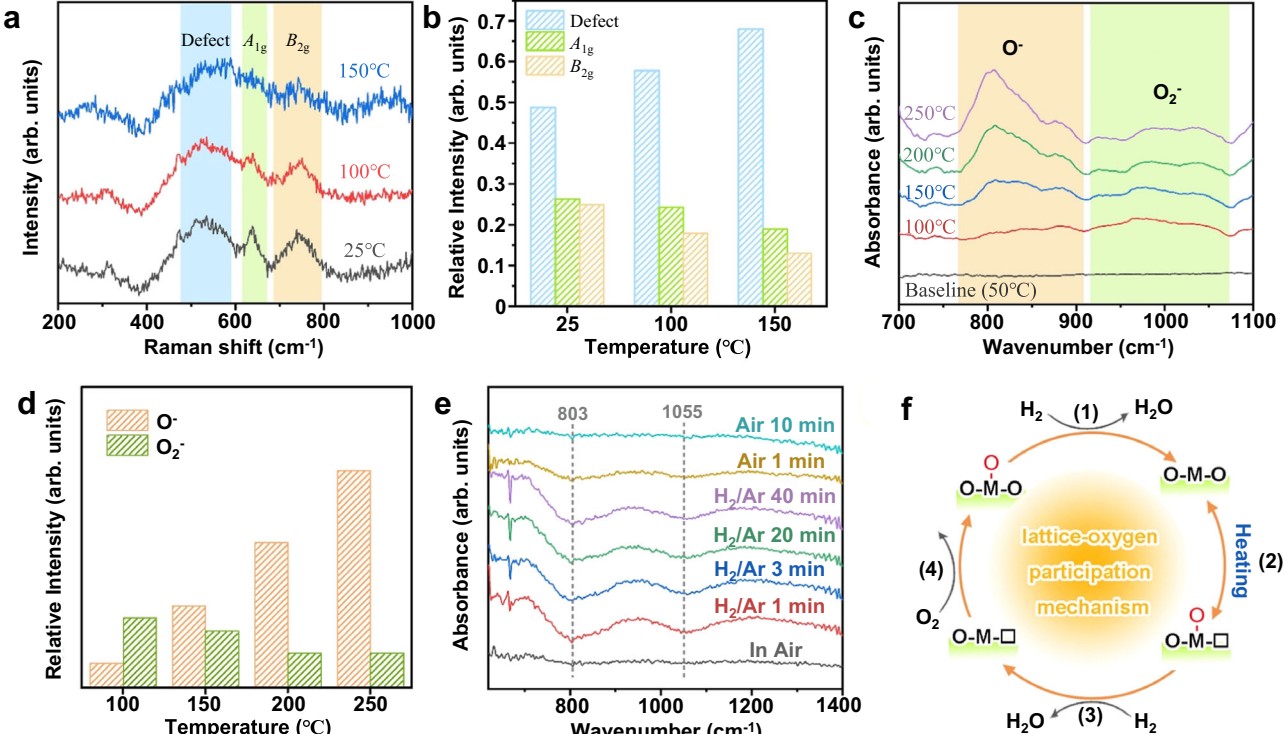

**Fig. 4 | Dynamic evolution of the gas-sensing reaction. a** In-situ Raman investigation for SGO at 25 °C, 100 °C and 150 °C, respectively. **b** Recorded in-situ Raman normalized intensity changes of Raman peaks at 535, 642, and 739 cm⁻¹ during heating processes. **c** In-situ DRIFTS investigation for SGO at 50–250 °C. **d** Recorded in-situ DRIFTS normalized intensity changes of $O_2^-$ and $O^-$'s DRIFTS peaks during heating processes. **e** In-situ DRIFTS investigation at 220 °C and continuous stream conditions in the subsequent 50 min, introduce air, H₂/Ar, air in turn. **f** Schematic diagram of the mechanism of the gas-sensing reaction with lattice oxygen

participation. (1) The chemisorbed oxygen reacts with hydrogen lead to a drop in resistance. (2) Surface lattice oxygen transforms into chemisorbed oxygen and generates surface oxygen vacancies. (3) The newly generated chemisorbed oxygen keeps on reacting with hydrogen and leads to a further drop in resistance. (4) The oxygen becomes adsorbed onto the gas-sensing material's surface and undergoes a conversion into surface lattice oxygen and chemically adsorbed oxygen. In all figures, DRIFTS denotes diffuse reflectance infrared Fourier transform spectroscopy.

We quantified the mass loss of SGO related to surface oxygen vacancy formation by thermogravimetric analysis. As depicted in Supplementary Fig. 28, SGO exhibits weight loss during heating, while the weight of SO remains relatively constant in a nitrogen environment. The weight loss of SGO is approximately 1.063% higher compared to SO at 220°C. We attribute this mass loss to the release of lattice oxygen. We estimated the depth of lattice-oxygen released from the SGO by combining with the grain size of SGO (detailed calculations are in the Supporting Information and Supplementary Fig. 29). The results of the calculations indicated that the thickness of lattice oxygen released is 0.045 nm. Based on the Sn-O bond length of approximately 0.2 nm, our findings indicate that only the surface lattice oxygen becomes partially revealed at an operating temperature of 220°C. Previous reports have demonstrated that deep lattice oxygen remains trapped, even at elevated temperatures[50].

Based on our investigation of the dynamic evolution of the SGO surface during the gas-sensing reaction, we have found evidence suggesting that the surface lattice oxygen actively participates in the hydrogen gas-sensing reaction. As a result, we have identified a reasonable mechanism for hydrogen-sensing reactions: the lattice oxygen participation mechanism (Fig. 4f). The mechanism involves a four-step process. First, after the hydrogen introducing, the chemisorbed oxygen reacts with hydrogen, and the electrons captured by the chemisorbed oxygen are released back into the gas-sensing material and lead to a drop of resistance. Next, during heating, surface lattice oxygen transforms into chemisorbed oxygen and generates surface oxygen vacancies on the lattice surface. In the third step, the newly generated chemisorbed oxygen keeps on reacting with hydrogen, and lead to a further drop in resistance. Ultimately, after reintroducing the sensor

into the atmosphere, the oxygen present in the air is adsorbed onto the gas-sensing material's surface and undergoes a conversion into surface lattice oxygen and chemically adsorbed oxygen.

Compared to the conventional hydrogen sensing mechanism, the lattice oxygen participation mechanism generates in-situ surface oxygen vacancies that enhance the response value of the gas sensing material. This is due to the conductive doping levels that are produced in the forbidden band region of the gas-sensing material because of the generation of surface oxygen vacancies[41,51,52]. These doping levels further reduce the resistance value ($R_g$) of the gas-sensing material in the target gas, leading to an increase in the response value ($S = R_a/R_g$).

To gain a deeper understanding of whether introducing Ge can promote the occurrence of LOM mechanism, we conducted DFT calculations. The model was created using a $2 \times 2 \times 2$ rutile SO unit cell, with four Ge atoms substituting four out of sixteen Sn atoms (Fig. 5a) (for more details, refer to the Methods and Supplementary Table 5). We investigated the changes in the crystal structure of SO after introducing Ge atoms. The lattice constant for SGO ($9.50 \text{ Å} \times 9.52 \text{ Å} \times 6.36 \text{ Å}$) is smaller than that of SO ($9.60 \text{ Å} \times 9.60 \text{ Å} \times 6.47 \text{ Å}$), which is consistent with experimental results. This compressive lattice strain leads to the compression and distortion of the SnO₆ octahedron. Consequently, the length of the Sn-O bond along the *a*-axis and *b*-axis is stretched from 2.08 Å to 2.10 Å, while the length along the *c*-axis is changed to 2.05 Å (Supplementary Fig. 30). Additionally, the bond angle of Sn-O-Sn, when viewed from the (110) direction, is altered from 180° to 162°, which maybe result in alterations to the electronic structure (Fig. 5b).

Then, the effect of introducing Ge atoms in SO on the valence charge was analysis. The electron local function (ELF) and the Bader

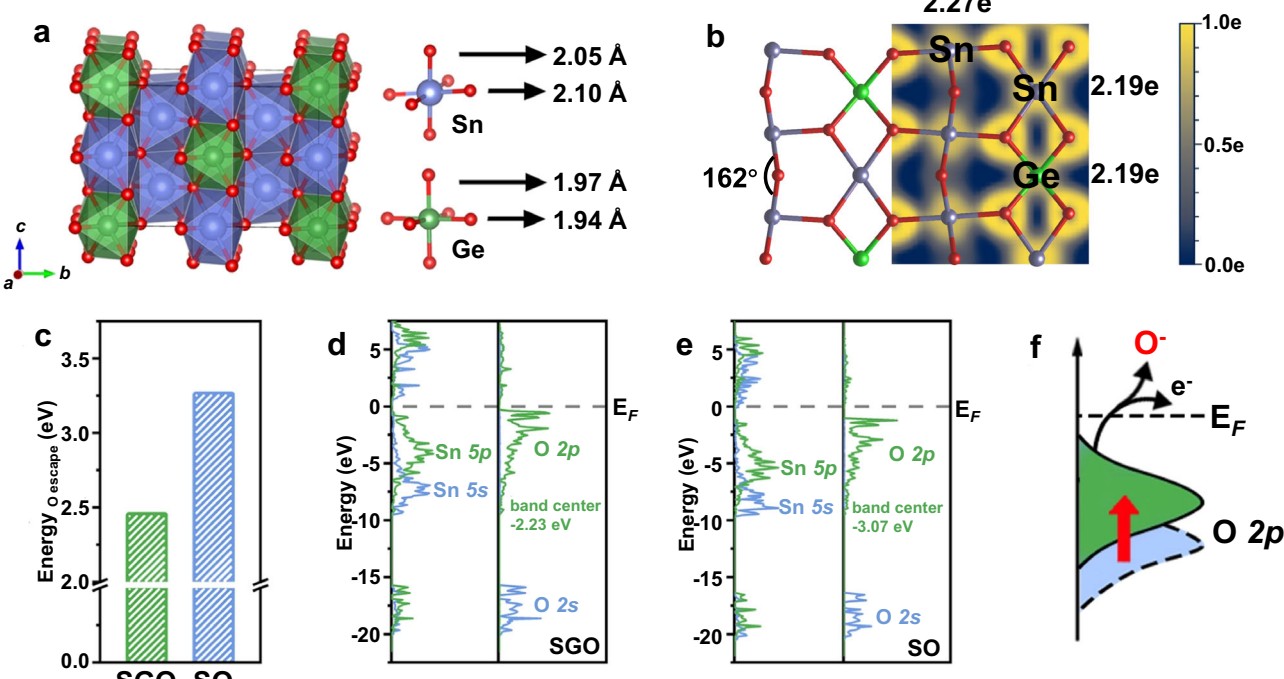

**Fig. 5 | DFT Calculation.** DFT Calculation. **a** Schematic illustration of SGO. The red, blue, and green atoms represent O, Sn, and Ge, respectively. **b** Electron local function (ELF) and the bader charge of SGO. **c** Escape energy of O atoms in SGO and SO. The electronic density of states (DOS) of SGO **d** and SO **e**, in which the Fermi level is set to 0. **f** The corresponding schematic illustration of the relationship between the position of O $p$-band (relative to Fermi level) and the conversion from surface lattice oxygen (O$^{2-}$) to chemisorbed oxygen (O$^-$). In all figures SGO denotes $Sn_{0.8}Ge_{0.2}O_2$ and SO denotes $SnO_2$. Blue/green color corresponds to Sn/Ge throughout the Figure c and f.

charge analysis were explored on the (110) surface of SGO. The results show that, compared with pure SO surface (2.19 |e| and 2.16 |e|), the charges on both of the two kinds of the Sn atoms in (110) surface are increased (2.27 |e| and 2.19 |e|), while there are no changes on the charges of O atoms in the surface (1.09 |e| vs 1.09 |e|) (Fig. 5b and Supplementary Fig. 31). This reveals that there is a complex electron transferring way in the SGO (Sn→O→Ge), which is agree with XPS of Sn.

Next, we investigated the alteration in the oxygen escape energy in SO following the introduction of Ge atoms and the factors contributing to this change. As shown in Fig. 5c, the oxygen escape energy in SGO is significantly lower than that of SO (2.45 eV vs 3.26 eV). This implies that the lattice oxygen are more easily transferred to form chemisorbed oxygen for SGO compared with SO. The electronic density of states for SGO and SO are calculated, which are shown in Fig. 5d and e, in order to investigate the impact of electronic structure on O escape energy. The peaks located around -20 eV are the hybridization of the O $2s$ and Sn $5s$, $5p$ orbital. This part is far from the fermi level, which means the electron located in these orbital is relatively inert for the chemical reaction. Around the Fermi level, the band are contributed mainly by O $2p$ orbital, accompanied by Sn $5p$ below. Moreover, SGO has a higher O $2p$-band center (relative to fermi level) than SO (−2.23 eV vs −3.07 eV), suggesting that the transformation from lattice oxygen to chemisorbed oxygen are thermodynamically more favorable of SGO (Fig. 5f). This further implies that introducing Ge and the resulting lattice distortion contribute to the optimizing of the electronic structure, as discussed in detail in the Supplementary Information (Supplementary Figs. 32 and 33). Consequently, during the gas sensing reaction, the surface lattice oxygen of SGO is more readily released and actively participated in the gas sensing process. This aligns with the results obtained from our in-situ spectroscopic characterization.

In summary, by combining in-situ characterization with DFT studies, we have demonstrated the lattice oxygen can transform into active chemisorbed oxygen species and participate in the sensing reaction, in addition to the classically studied mechanism on semiconductor metal oxides. Moreover, the reactivity of lattice oxygen can be triggered by the location of the O $p$-band center. Specifically, the upward shift of the O $p$-band center is beneficial to the escape of surface lattice oxygen, rendering the enhanced performance of $H_2$ sensing in Ge-doped SO. Our research has contributed to the advancements in designing hydrogen-sensing materials and understanding gas-sensing mechanisms.

## Methods
### Materials synthesis
All chemical reagents used in this study were commercial purchases and were used without further purification. The SGO nanofibers and SO nanofibers were all synthesized by electrospinning in this experiment. Firstly, Tin(IV) tetrachloride pentahydrate (0.618 mmol), N,N-Dimethylformamide (2.2 g) and ethanol (6.6 g) were mixed with stirring. Then 0 mmol, 0.069 mmol, 0.155 mmol, 0.265 mmol or 0.412 mmol of bis(2-carboxyethyl germanium(IV) sesquioxide) (Ge-132) was added to the above solution and dissolved with stirring. Last polyvinylpyrrolidone (0.8 g) was put into the solution and stirred for 12 h until the solution was transparent. Transfer the solution to a 5 ml medical syringe. Fix a piece of aluminum foil as a receiver at a distance of 20 cm from the tip of the syringe. A voltage of 22 kV is applied between the needle tip and the receiver. During the electrospinning process, keep the relative humidity and the temperature in the electrospinning chamber at 20%~30% and 20~25 °C, respectively. The obtained fibers were heated at 600 °C for 2 h.

### Characterizations
Scanning electron microscopy (SEM) was carried out on a JEOL JSM 6700 F electron microscope (Japan). The nitrogen absorption and desorption investigation were obtained on a Micromeritics model. The

powder X-ray diffraction (XRD) patterns were obtained by a Rigaku D/Max 2550 X-ray diffractometer (Japan) using Cu Kα radiation ($\lambda = 1.5418$ Å). X-ray photoelectron spectroscopy (XPS) were carried out on an ESCALAB 250 X-ray photoelectron spectrometer with a monochromatic X-ray source (Al Kα hμ = 1486.6 eV) (USA). Raman was carried out on a Renishaw Raman system model 1000 spectrometer (UK) by using a 532 nm excitation source. Transmission electron microscopy (TEM), high-resolution TEM (HRTEM) images, and energy-dispersive X-ray spectroscopy (EDS) spectra were obtained by a Philips-FEI Tecnai G2S-Twin equipped with a field emission gun operating at 200 kV (Netherlands). UV-Vis diffuse reflectance analyzes were performed on a PerkinElmer Lambda 20 UV-vis spectrometer (USA). Work functions of the as-obtained materials were obtained on a scanning Kelvin probe system (KP Technology Ltd) in the air (UK). $O_2$ temperature-programmed desorption ($O_2$-TPD) analyzes were performed on a Micromeritics AutoChem 2920 II system (USA). ASAP 2020 M system (USA).

## Fabrication and measurement of the gas sensor

A matching base, a nickel-chromium heating wire, a ceramic tube with two Au electrodes, two Pt wires at each Au electrode, and a sensing layer covered on the tube made up the gas sensor. To make a sensor, the sensing material was mixed with ethanol to make a viscous slurry, which was then evenly coated over the ceramic tube's surface. The current through the heating wire was used to control the sensor's operational temperature. Before the sensing test, the as-fabricated sensors were aged at 160 °C in the air for 24 hours to increase their stability and repeatability. A commercial CGS-8 gas sensing measuring system (Beijing Elite Tech Co., Ltd China) was used for gas sensing testing. The volume of the test chamber is 2.5 L. In order to prepare a homogeneous target gas with the desired concentration, a specific number of $H_2$ gas or the interference of other gases were introduced into the test chamber and retained for at least 30 minutes. The sensor was placed in a chamber filled with fresh air to investigate its recovery properties. The sensing experiments were carried out at temperatures ranging from 160 to 260 °C. The concentrations of $H_2$ ranged from 0.05 to 2000 ppm. The time it took the sensor to accomplish (recover to) 90% of the total resistance change was defined as the response time (recovery time). $S = R_a/R_g$ was defined as the sensor response value, where $R_g$ and $R_a$ are sensor resistances in target gas with specific concentrations and air, respectively.

## Test details of in-situ DRIFTS and in-situ Raman

In-situ DRIFTS measurements were carried out using the Bruker IFS 66 V/S FTIR spectrometer (USA), which was equipped with an in-situ diffuse-reflectance reaction chamber (Supplementary Fig. 27). We first investigated the effect of temperature on the adsorbed oxygen on the surface of CGO using in-situ DRIFTS techniques. Initially, argon (Ar) was used to purge the gas lines and reaction chamber. Then, dry air was introduced into the chamber, and the sample within the chamber was heated in a stepwise manner from 50 °C to 250 °C, with each increment of 50 °C.

Subsequently, we examined the impact of introducing $H_2$ at the optimal working temperature of SGO (220 °C) on the surface adsorbed oxygen using in-situ DRIFTS. The test condition of in-situ DRIFTS was very similar to the atmosphere of $H_2$ gas-sensing reaction. Initially, the test chamber was heated to 220 °C. Subsequently, dry air, followed by a mixture of 5% $H_2$ and Ar, and finally dry air again, were sequentially purged into the chamber.

In-situ Raman measurements were carried out using the Renishaw Raman system model 1000 spectrometer (UK), by using an air-cooled argon ion laser (532 nm) as the excitation source, which was equipped with an in-situ reaction chamber. We investigated the effect of temperature, from 25 °C to 150 °C, on the crystal structure of CGO using in-situ Raman techniques.

## Computation details

All DFT calculations are carried out in the frame of Vienna ab initio simulation package (VASP5.4.4)[53,54]. The Perdew-Burke-Ernzerhof (PBE) exchange correlation functional in the form of Generalized gradient approximation (GGA) is employed to describe the exchange-correlation potential[55]. The project augmented wave method is used to evaluate the electron-ion interaction and the plane wave cut-off energy is set as 450 eV[56]. The threshold criteria are settled as $10^{-4}$ eV in energy and 0.05 eV/Å in force, respectively. The Morkhorst-pack method is used to sample the brillizone and the separation length for k-point mesh is set as 0.03 $2\pi$ Å$^{-1}$ for optimization process[57]. For density of states (DOS) calculations, a 0.02 $2\pi$ Å$^{-1}$ length is chosen. The k-points file is generated automatically using VASKIT Code[58]. For surface models, the long-range van der Waals (vdW) correction is taken into account using DFT-D3 method proposed by Grimme[59,60]. During the surface calculation process, the symmetrisation is switched off, and the dipole correction was adopted.

## Theoretical model of SGO

The model of SGO solid solution was constructed through the following three steps. First, a rutile $SnO_2$ structure was used as the initial model. Next, the unit cell was enlarged to $2 \times 2 \times 2$ supercell containing 16 Sn atoms. In our study, we substituted 4 Sn atoms with 4 Ge atoms. The substitution was done by selecting the most stable structure at each ratio (e.g., 15:1, 14:2, 13:3, 12:4) in a sequential manner. The structure with the lowest energy was always chosen as the initial structure for the next proportional replacement. The various configures of the doping situations were shown in Supplementary Table 5, and the structure with the lowest energy for each ratio was marked in red and chosen as the SGO.

The (110) surface of SGO was chosen to further explore the properties SGO as the (110) surface is the most stable surface in rutile structures[61]. A vacuum layer with 15 Å thickness is inserted into two neighbor slabs. The slab model of SGO contains three Sn(Ge)$O_6$ octahedron layers and half of the bottom atomic layer was frozen to simulate the bulk phase, while the remaining atoms in the model are fully relaxed. The (110) surface model of $SnO_2$ was obtained from the same method.

## Computation methods of O-2p band center and the escape energy of the surface O atom

The O-2p band center ($\varepsilon_{2p}$) was calculated using the following Eq. (1)[62]:

$$\varepsilon_{2p} = \frac{\int_{-\infty}^{\infty} n_p(\varepsilon)\varepsilon d\varepsilon}{\int_{-\infty}^{\infty} n_p(\varepsilon) d\varepsilon} \tag{1}$$

in which $\varepsilon$ is the energy relative to the Fermi level, the $n_p(\varepsilon)$ is the projected DOS data of the surface O atoms.

The escape energy of the surface O atom was assumed as the following process and the correlative free energy for this special reaction was evaluated using the following Eqs. (2–3)[63]:

$$SGO\_O \rightarrow SGO\_O_e + \frac{1}{2}O_2 \tag{2}$$

$$\Delta G = E_{SGO\_O_v} - E_{SGO\_O_{sur}} + \frac{1}{2}G_{O_2} \tag{3}$$

In which SGO_O is the initial surface, and SGO_Oe is the SGO surface with an O escaping. And $E_{SGO\_Oe}$, $E_{SGO\_Osur}$ are the energies of their theoretical models, respectively.

## Reporting summary

Further information on research design is available in the Nature Portfolio Reporting Summary linked to this article.

## Data availability

The data that support the findings of this study are available within the article and its Supplementary Information. The Source data generated in this study have been deposited in the Figshare database[64]. Source data are provided with this paper.

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

## Acknowledgements

G.L., R.G. and Q.L. acknowledge funding from the National Natural Science Foundation of China (NSFC) (No. 21975093 to G.L. and No. 22209205 to Q.L.), the Scientific Research Fund of Zhejiang Provincial Education Department (No. Y202249731 to R.G.) Natural Science Foun-dation of Zhejiang Province of China (LQ24B060010 to R.G.) and Open Research Fund of State Key Laboratory of Inorganic Synthesis and Pre-parative Chemistry (Jilin University, No. 202326 to R.G.).

## Author contributions

G.L. directed this research. J.L. conducted most of the experiments. L.S., W.A., Z.Z. and L.Z. performed the theoretical calculations. N.B. and X.Z. contributed to data analysis. J.L. and Q.L. wrote the paper. R.G. and G.L. supervised the project. All authors discussed and reviewed the final manuscript.

## Competing interests

The authors declare no competing interests.
