## [Peer Review File · Nature Communications]

Essential role of lattice oxygen in hydrogen sensing reactionREVIEWER COMMENTS

Reviewer #1 (Remarks to the Author):

In the submitted manuscript, the authors demonstrated essential role of lattice oxygen in hydrogen sensing reaction. The researchers used in-situ Raman and diffuse reflectance infrared Fourier transform spectroscopy (DRIFT) to explore the hydrogen-sensing mechanism, proposing a novel mechanism involving lattice oxygen participation. However, before publication in Nature Communications and share to wider readership, several questions related to this work should be addressed. So far, I do not agree to this article being published in Nature Communications.

1. There are some typos and grammar errors; the English sentence needs further improvement. (page 2 : H₂, page 11 : deceased etc.)

2. Page 7, line 140-When the operating temperature of the sensor is increased, it promotes the reaction with hydrogen, leading to an increase in sensitivity. It was stated that exceeding the optimal temperature results in a decrease in sensitivity due to the reduction of active sites. However, if the active site, in this case, the SGO lattice oxygen, is abundant, shouldn't the sensitivity remain high? The reason for the sensitivity decreasing again is unclear.

3. I am confused about how the shape of the response-recovery time graph for 200 ppb is modified in Page 8, Figure 2d.

4. Page 7. What does SAE stand for?

5. Page 8. The description for $S = 39.2$ for H₂ in the gas selectivity graph is different from the graph. There is also no indication of H₂ concentration.

6. Page 12. The analysis confirming the nature of O- adsorption at 220 degrees operating temperature or the absence of references is not clear.

7. In the DFT calculation model of Figure 5, the formation energy of oxygen vacancy due to Ge doping was not considered. The model is simplistic.

8. In Figure 4f, "heating" of the lattice-oxygen participation mechanism is unclear exactly what atmosphere the heat treatment is in. In Figure S16, we estimated the atmosphere of "heating" from the comparison between Ar and air atmosphere. In this regard, the most crucial mechanism of this study is the transformation of lattice oxygen into chemisorbed oxygen in an inert atmosphere. Therefore, we recommended that this research would be a more fundamental study if it also addresses whether the tendencies of the dynamic response graph in argon and air are the same for all other gases that react with chemisorbed oxygen or if not, the reasons for any reason.

9. The formation of oxygen vacancies results in a mass loss of the oxide support. We suggest conducting additional analyses, such as thermogravimetric analysis (TGA), to quantify the mass loss related to oxygen vacancy formation and provide further evidence for your proposed mechanism.

10. Additional experimental details on gas behavior in an inert atmosphere are needed. According to the claimed mechanism, lattice oxygen-derived chemisorbed oxygen formed in the heating step will continue to be formed even when hydrogen is injected. Still, I am curious about the reason why the graph was saturated after the reaction.

The driving mechanism utilizing adsorbed oxygen species in metal oxide-based chemiresistive sensors has been theoretically established and refined based on the operating temperature. This paper aimed to propose a reaction mechanism where surface lattice oxygen acts as chemisorbed oxygen at 220 degree, deviating from these established theories. However, the in-depth analysis and sensor measurement results related to this proposal appear to be quite inadequate. The sensor results related to this are solely presented in Figure S16. The transition and flow of content from Figure 1, 2, 3 to Figure 4, 5 in the subsequent sections appear quite awkward. Therefore, I believe it might be somewhat challenging for this paper to be accepted in the Nature Communications.

Reviewer #2 (Remarks to the Author):

The authors report a hydrogen sensitive SGO and the sensing mechanism. It is well-structured, and the findings presented in research are insightful and substantial. However, some points should be clearly addressed in the revised manuscript before publication. It is recommended to accept it after minor revision.

1. Please use EPR to determine the state of oxygen vacancies in SGO after gas sensing reaction.
2. In order to further verify the structural stability of SGO in the sensing reaction, please provide the post-reaction XRD, SEM, and other data of SGO.
3. In Figure 3f, two peaks are present in the O₂-TPD of SGO, while only one is present in SO, please analyze it in more detail.
4. In line 159, the authors inferred "the response time of SGO is significantly shorter compared to most hydrogen sensing materials, including those decorated with precious metals". Some related references should be cited.
5. In the DFT portion, the authors should elucidate whether lattice distortion or Ge electronic interaction results in the upshift of the p band center in SnO₂.
6. The authors built a model with 25% Ge doping in SnO₂ and observed an upshift in the p band center. It would be interesting to investigate how the ratios of Ge doping affect these characteristics in SnO₂.
7. Can the lattice oxygen participate in other gas-sensing processes, such as formaldehyde, benzenes or other VOC?
8. There are some formatting errors in the text, please check it carefully.

Reviewer #3 (Remarks to the Author):

The authors describe sensing results to Ge-doped SnO₂ samples, attempting to rationalise the higher performance using XPS, DRIFTS and DFT calculations. However there are deficiencies in their analysis that mean I am unconvinced by some of their data, and some of their rationalisation appears to be pure speculation. They have also missed some key literature that would likely allow them to provide a more convincing discussion. Consequently I am recommending rejection of the article, and provide the detailed basis for this below (as well as notes of more general corrections), which hopefully will help the authors amend the article for future publication elsewhere.

Line (L)40; pollution from hydrogen combustion is entirely dependent on how it was produced

L42; one assumes that is the explosive concentration mixed with air?

L53-57; this description is largely superfluous as it is repeated later

L78; better to refer to Ge-132 using its chemical name or formula

L83; That shift would only be relevant to the peak shown at around 26 degrees unless it is the result of a systematic error. If the unit cell has changed size then the relevant lattice parameters should be given for both rutile-SO, SGO and for a reference of SO

L86/87; the evidence for this should be shown (at least included in SI)

L97/98; Not convinced by this - the apparently higher intensity in this region compared to SO appears more likely to be a function of the weak intensity of the A1g and B2g bands rather than an increase in intensity of this peak

L110-112; Need to demonstrate that this shift is not observed in, e.g. C1s or O1s peaks as well (see later point also)

L122/123; Is this consistent with the reduced lattice size of SGO compared to SO - comparison with equivalent HRTEM of SO would be useful

L140-142; Where is the evidence for this?

L151/152; What is beta?

L208/209; By eye the shift in binding energy of Sn 3d of around 0.4 eV in Figure 1c. If so, this

change in Fermi energy would account for the change in Sn peak position, not any change in the valence state of Sn as suggested above. Comparing Figure 3e with S15, I am pretty sure there is a shift on the O1s also, which indicates a change in Fermi level and not a change in valence state of Sn.

L217-219; This ignores the active site and the fractional occupation of that active site with oxygen. Also, it would be relatively easy to evidence this conjecture by looking at the resistance before and after H₂ exposure for SO and SGO. It is always worth including raw resistance data in SI, not just response data.

L224-226; the idea that XPS can show any information regarding oxygen vacancies, in particular for any samples that have been air exposed, has been comprehensively debunked (10.1016/j.susc.2021.121894 and 10.1021/acs.chemmater.3c00801). Consequently the entire section on XPS needs revisiting.

L247/248; This seems to directly contradict the authors statements above using XPS data

L293-297; the mechanism proposed is very strange. It seems to be a hybrid between a traditional gas sensing mechanism using ionosorbed oxygen and the Mars van Krevelen mechanism of oxidation used in catalysis. I am not sure why the authors want there to be charged chemical species on the surface, when their evidence is all about the oxygen vacancy concentration (e.g. EPR) – I am not expert in DRIFTS but even a quick search revealed the peak around 1050 is associated with Sn-O and therefore I see no need to invoke species they have not measured. There are several examples in literature of sensing mechanisms that are discussed only in terms of lattice oxygen/oxygen vacancies, e.g. 10.1016/0040-6090(88)90424-5, 10.1016/0167-2584(87)90918-2, 10.1021/acs.jpcc.8b01446, 10.1021/acssensors.1c01727

L304-306; most recent calculations on SnO₂ suggest that bulk oxygen vacancies are too deep to be significantly ionised, even at sensor operation temperatures - 10.1103/PhysRevMaterials.2.054604

L309-312; this requires far more discussion

RESPONSE TO REVIEWERS' COMMENTS

Reviewer #1 (Remarks to the Author):

In the submitted manuscript, the authors demonstrated essential role of lattice oxygen in hydrogen sensing reaction. The researchers used in-situ Raman and diffuse reflectance infrared Fourier transform spectroscopy (DRIFT) to explore the hydrogen-sensing mechanism, proposing a novel mechanism involving lattice oxygen participation. However, before publication in Nature Communications and share to wider readership, several questions related to this work should be addressed. So far, I do not agree to this article being published in Nature Communications.

Response: We appreciate the reviewer's detailed reading and pertinent advice to our work. Below, we tried to answer all the raised comments.

1. There are some typos and grammar errors; the English sentence needs further improvement. (page 2 : H₂, page 11 : deceased etc.)

Response: Thanks. Typos, grammar errors and English sentences have been carefully checked and improved.

2. Page 7, line 140-When the operating temperature of the sensor is increased, it promotes the reaction with hydrogen, leading to an increase in sensitivity. It was stated that exceeding the optimal temperature results in a decrease in sensitivity due to the reduction of active sites. However, if the active site, in this case, the SGO lattice oxygen, is abundant, shouldn't the sensitivity remain high? The reason for the sensitivity decreasing again is unclear.

Response: We appreciate this important comment. We are sorry that our expression is not rigorous and makes the confusion. First, increasing temperatures can not cause a decrease in the number of active sites available for H₂ adsorption. Second, for the sensing process, the operating temperature was a key parameter for gas sensing performance. When the operating temperature was changed, both the adsorption-desorption procedure and redox reaction rate would be greatly affected. With the increase in the operating temperature, the hydrogen molecules gain enough energy to overcome the activation energy barrier to react with the oxygen species. However, when the temperature is too high, gas molecules move faster and desorb before they have time to participate in the gas-sensing reaction. Hence the responses of SGO exhibit a trend of "increase-maximum-decrease" as the operating temperature increases.

Revision: We have modified the "This is due to the promotion of surface reaction rates at higher temperatures, but also a decrease in the number of active sites available for H₂ adsorption at high temperatures." into "With the increase in the operating temperature, the hydrogen molecules gain enough energy to overcome the activation energy barrier to react with the oxygen species. However, when the temperature is too high, gas molecules move faster and desorb before they have time to participate in the gas-sensing reaction. Hence, the responses of SGO exhibit a trend of "increase-maximum-decrease" as the operating temperature increases."

3. I am confused about how the shape of the response-recovery time graph for 200 ppb is modified in Page 8, Figure 2d.

Response: Sorry for the carelessness. The response-recovery curve was recorded with 1000 ppm of H₂ in **Figure 2d**. In the manuscript, the value we wrote was 1000 ppm, but in the legend of **Figure**

2d, we mistakenly wrote it as 200 ppb. We have made the necessary corrections.

4. Page 7. What does SAE stand for?

Response: Sorry for the carelessness. SAE stand for Society of Automotive Engineers and we have added it in the manuscript.

5. Page 8. The description for $S = 39.2$ for H_2 in the gas selectivity graph is different from the graph. There is also no indication of H_2 concentration.

Response: Sorry for the carelessness. There is systematic error and random error during measurement (less than 5%). We have performed repeated tests on SGO with 100 ppm H_2 and added an error bar in **Figure 2e**. It can be observed that $S = 39.2$ is also included in this range. We have modified the sentences in the revised version as follows: “Specifically, the sensor exhibits a response value of $S = 37.6 \pm 1.92$ for 100 ppm H_2 , which is at least seven times greater than any of the interfering gases.”

Rev. Figure 2e. Selectivity to H_2 in the presence of CH_4 , CO , ethane, propane, butane and CO_2 .

6. Page 12. The analysis confirming the nature of O^- adsorption at 220 degrees operating temperature or the absence of references is not clear.

Response: Thanks for the reminder! In previous report (10.1023/A:1014405811371; 10.1021/acscatal.9b02408), the researchers analyzed the species of O^- above $150^\circ C$ by TPD, FTIR, and ESR, they confirmed O^- is the main species on the surface of oxides. Furthermore, we have added relevant references.

7. In the DFT calculation model of Figure 5, the formation energy of oxygen vacancy due to Ge doping was not considered. The model is simplistic.

Response: We appreciate the insightful comments. We initially overlooked the significance of the formation energy of oxygen vacancies (to express it more appropriately, we named it the escape energy of O atoms, See **Methods** for calculation details) and included it in the SI (quodam **Supplementary Figure 21**). Taking your suggestions into account, we have now relocated it to **Figure 5c** of the manuscript. Additionally, we have supplemented our modeling process, as detailed in **Supplementary Table 4**.

Revision: We added modeling details in the **Methods** section of the manuscript. “In our study, we substituted 4 Sn atoms with 4 Ge atoms. The substitution was done by selecting the most stable

structure at each ratio (e.g., 15:1, 14:2, 13:3, 12:4) in a sequential manner. The structure with the lowest energy was always chosen as the initial structure for the next proportional replacement. The various configurations of the doping situations were shown in Supplementary Table 4, and the structure with the lowest energy for each ratio was marked in red and chosen as the SGO.”

Rev. Figure 5. DFT Calculation. (a) Schematic illustration of SGO. (b) Electron local function (ELF) and the Bader charge of SGO. (c) Escape energy of O atoms in SGO and SO. The electronic density of states (DOS) of SGO (d) and SO (e). (f) Schematic representation of the energy level versus density of states, showing the respective motion of the oxygen p -band with respect to the Fermi level in going from surface lattice oxygen (O^{2-}) to chemisorbed oxygen (O).

Rev. Supplementary Table 4. The possible bulk structures of SGO with germanium content from 6.25% to 25%, and the geometric optimization of total energies in DFT calculations. The model with the lowest total energy in each germanium content (red marked) was chosen as the corresponding SGO bulk model.

Sn: Ge	 Sn	 Ge	 O
1:0	 $E_{total} = -300.52 \text{ eV}$		
15:1	 $E_{total} = -300.52 \text{ eV}$		
14:2	 $E_{total} = -300.56 \text{ eV}$	 $E_{total} = -300.38 \text{ eV}$	 $E_{total} = -300.67 \text{ eV}$
	 $E_{total} = -300.62 \text{ eV}$	 $E_{total} = -300.42 \text{ eV}$	 $E_{total} = -300.50 \text{ eV}$
15:3	 $E_{total} = -300.75 \text{ eV}$	 $E_{total} = -300.64 \text{ eV}$	 $E_{total} = -300.74 \text{ eV}$
	 $E_{total} = -300.76 \text{ eV}$	 $E_{total} = -300.18 \text{ eV}$	
12:4	 $E_{total} = -300.88 \text{ eV}$	 $E_{total} = -300.68 \text{ eV}$	 $E_{total} = -300.63 \text{ eV}$
	 $E_{total} = -301.01 \text{ eV}$	 $E_{total} = -300.94 \text{ eV}$	 $E_{total} = -300.89 \text{ eV}$

8. In Figure 4f, “heating” of the lattice-oxygen participation mechanism is unclear exactly what atmosphere the heat treatment is in. In Figure S16, we estimated the atmosphere of “heating” from the comparison between Ar and air atmosphere. In this regard, the most crucial mechanism of this study is the transformation of lattice oxygen into chemisorbed oxygen in an inert atmosphere. Therefore, we recommended that this research would be a more fundamental study if it also addresses whether the tendencies of the dynamic response graph in argon and air are the same for all other gases that react with chemisorbed oxygen or if not, the reasons for any reason.

Response: Thank you for your suggestion, it is very helpful to improve our work. We conducted experiments to investigate responses of SGO to the gases that can react with adsorbed oxygen under both air and argon conditions, such as CO, formaldehyde (FA) and ethanol (ET). The results (**Supplementary Figure 25**) indicate that the response values of CO, FA and ET in the argon environment are improved compared to those in the air environment, at operating temperature of 220°C. Therefore, we deduced that the conversion of lattice oxygen to chemisorbed oxygen may be a ubiquitous phenomenon at an operating temperature of 220°C.

Rev. Supplementary Figure 25. The response values of SGO to H₂, CO, formaldehyde, ethanol in air and argon, respectively.

9. The formation of oxygen vacancies results in a mass loss of the oxide support. We suggest conducting additional analyses, such as thermogravimetric analysis (TGA), to quantify the mass loss related to oxygen vacancy formation and provide further evidence for your proposed mechanism.

Response: Thank you for your suggestion, which greatly improves our work. We have incorporated thermogravimetric analysis into the revised manuscript. As depicted in **Supplementary Figure 27**, SGO exhibits weight loss during heating, while the weight of SO remains relatively constant in nitrogen environment. The weight loss of SGO is approximately 1.063% compared to SO, at 220°C. We attribute this additional mass loss to the release of lattice oxygen.

In order to estimate the thickness of lattice-oxygen released from the SGO (x in **Supplementary Figure 28**), we assume that 1) the particle is regarded as a regular sphere, and the diameter of this sphere is set as 5.6 nm according to the size distribution diagram (Supplementary Figure 6), 2) the O atom is assumed distributed uniformly in the particle. Then, the percent of O escape can be corresponding to the ratio between the O escape volume and the whole particle

volume, which is described as:

Oxygen ratio_{released} = $4/3 \times \pi r_1^3 - 4/3 \times \pi r_2^3$. Considering the experimental results that the escaping percent of O% is about 1.063%, and the value of R is half of diameter, which is 2.8 nm, the value of x can be obtained from the above equation $x = 0.045 \text{ nm} = 0.45 \text{ \AA}$.

Based on the Sn-O bond length of approximately 0.2 nm, our findings indicate that only the surface lattice oxygen becomes partially revealed, at an operating temperature of 220°C.

Revision: We have added relevant results to the manuscript. “We quantified the mass loss of SGO related to surface oxygen vacancy formation by thermogravimetric analysis. As depicted in **Supplementary Figure 27**, SGO exhibits weight loss during heating, while the weight of SO remains relatively constant in nitrogen environment. The weight loss of SGO is approximately 1.063% compared to SO, at 220°C. We attribute this mass loss to the release of lattice oxygen. We approximated the depth of lattice-oxygen released from the SGO, by combining with the grain size of SGO (detailed calculations are in the **Supplementary Figure 28**). The results of the calculations indicated that the thickness of lattice oxygen released is 0.045 nm. Based on the Sn-O bond length of approximately 0.2 nm, our findings indicate that only the surface lattice oxygen becomes partially revealed, at an operating temperature of 220°C. Previous reports have demonstrated that deep lattice oxygen remains trapped, even at elevated temperatures.”

Rev. Supplementary Figure 27. TGA plots of SGO and SO in nitrogen.

Rev. Supplementary Figure 28. The geometry simulation model for SGO particle.

10. Additional experimental details on gas behavior in an inert atmosphere are needed. According to the claimed mechanism, lattice oxygen-derived chemisorbed oxygen formed in the heating step will continue to be formed even when hydrogen is injected. Still, I am curious about the reason why the graph was saturated after the reaction.

Response: We appreciate the insightful comments. The gas-sensing reaction occurs on the surface of the oxides. That is to say, only the surface oxygen species can react with the target gas. As analyzed in Comment 9, the surface lattice oxygen can only partially transfer into absorbed oxygen while the deep lattice oxygen remains uninvolved at an operating temperature of 220°C. Hence, it is reasonable that saturated point appeared in the graph. To further verify above analysis, we further performed H₂-TPR analysis on SGO using a 10%H₂/90%Ar mixture. The signal intensity represents the amount of oxygen that participated in the gas-sensing reaction (**Supplementary Figure 14**). As shown by the black line in figure, in which the SGO are heated to 600°C at a rate of 10°C/min, the higher the operating temperature, the higher amount of oxygen that can take part in sensing reaction. Hence, the amount of oxygen that can participated in the sensing reaction has direct relationship with the operating temperature. When heating to 220°C at the same rate and maintaining this temperature (red line), the hydrogen reduction peak first rises and then gradually diminishes and eventually disappears. This further confirms that only a small amount of oxygen has the ability to react with hydrogen. In addition, the viewpoint that deep lattice oxygen does not participate in the reaction is also supported by relevant references (10.1103/PhysRevMaterials.2.054604).

Rev. Supplementary Figure 14. H₂-TPR profiles of SGO, red line indicates a heating process where the temperature is gradually increased to 220°C at a rate of 10°C/min, and then maintained at 220°C. On the other hand, the black line represents a heating process where the temperature is directly raised to 600°C at a rate of 10°C/min.

11. The driving mechanism utilizing adsorbed oxygen species in metal oxide-based chemiresistive sensors has been theoretically established and refined based on the operating temperature. This paper aimed to propose a reaction mechanism where surface lattice oxygen acts as chemisorbed oxygen at 220 degree, deviating from these established theories. However, the in-depth analysis and sensor measurement results related to this proposal appear to be quite inadequate. The sensor results related to this are solely presented in Figure S16. The transition and flow of content from Figure 1, 2, 3 to Figure 4, 5 in the subsequent sections appear quite awkward. Therefore, I believe it might be somewhat challenging for this paper to be accepted in the Nature Communications.

Response: Thank you very much for your valuable comments, which significantly enhanced the quality of our manuscript. We have also incorporated additional characterization and testing, such as TGA, H₂-TPR, the response values of SGO to CO and formaldehyde under both air and argon conditions. These new results provide further evidence of the reliability and applicability of our proposed mechanism. Furthermore, we further modified the logic of Figures 1, 2, 3 to Figure 4, 5 and added transition content.

Reviewer #2 (Remarks to the Author):

The authors report a hydrogen sensitive SGO and the sensing mechanism. It is well-structured, and

the findings presented in research are insightful and substantial. However, some points should be clearly addressed in the revised manuscript before publication. It is recommended to accept it after minor revision.

Response: We appreciate the reviewer's recognition of our effort to a hydrogen sensing SGO and the sensing mechanism. Below, we tried to answer all the raised comments.

1. Please using EPR to determine the state of oxygen vacancies in SGO after gas sensing reaction.

Response: Thank you for your valuable suggestions, which will contribute to the improvement of our work. We have provided EPR spectra of SGO before and after the gas sensing reaction. As depicted in **Supplementary Figure 16**, the signal from the oxygen vacancy remains relatively unchanged. This observation further supports the notion of the stability of SGO before and after gas-sensing reactions.

Rev. Supplementary Figure 16. EPR spectra of SGO before and after the gas sensing reaction.

2. In order to further verify the structural stability of SGO in the sensing reaction, please provide the post-reaction XRD, SEM, and other data of SGO.

Response: We appreciate your valuable suggestions, which are very helpful in improving the quality of our articles! We conducted XRD, SEM, TEM, and HRTEM analysis of SGO after the gas-sensitization reaction, as shown in **Supplementary Figure 15**.

Revision: We have added relevant characterization results to the manuscript. “The XRD patterns (**Supplementary Figure 15a**) before and after the reaction showed almost no change. The SEM image (**Supplementary Figure 15b**) of SGO after the reaction also showed no significant changes compared to pristine sample. The HRTEM image (**Supplementary Figure 15c**) of SGO after the gas-sensitized reaction revealed a clear lattice structure, indicating there is no surface amorphization. Therefore, it can be concluded that SGO maintains its crystal structure during the gas-sensing reaction.”

Rev. Supplementary Figure 15. (a) XRD patterns of SGO before sensing reaction and after sensing reaction. (b) SEM image of SGO after sensing reaction. (c) HRTEM image of SGO after sensing reaction.

3. In Figure 3f, two peaks are present in the O₂-TPD of SGO, while only one is present in SO, please analyze it in more detail.

Response: We greatly appreciate the valuable comments. In **Figure 3f**, we have provided a comprehensive attribution of the O₂-TPD peaks of SGO and SO by referring to relevant literature. According to the literature, the oxygen desorption peaks of SO and SGO can be categorized based on temperature ranges as follows:

- (1) 100–200°C: desorption of surface adsorbed peroxy species O₂⁻ (ad).
- (2) 200–400°C: desorption of surface adsorbed monatomic species O⁻ (ad).
- (3) 400–700°C: desorption of lattice oxygen O²⁻ (lattice).

Revision: We have modified the “As shown in **Figure 3f**, O₂-TPD demonstrates that SGO has higher chemisorbed oxygen compared to SO.” into “O₂-TPD profiles of SGO and SO (**Figure 3f**), the green line representing SO shows a weak O⁻ (ad) desorption peak between 300-400°C. On the other hand, the blue line representing SGO exhibits a very strong desorption peak. This indicates that there are considerably more chemisorbed monatomic species O⁻ on the surface of SGO compared to SO. This conclusion is supported by the XPS data of the O 1s.”

4. In line 159, the authors inferred “the response time of SGO is significantly shorter compared to most hydrogen sensing materials, including those decorated with precious metals”. Some related references should be cited.

Response: We greatly appreciate your valuable advice! The related references have been included in Supporting Information and their comparison with our work have been provided in Supplementary Table 2.

5. In DFT portion, the authors should elucidate whether lattice distortion or Ge electronic interaction results in the upshift of the p band center in SnO₂.

Response: Thank you for your valuable advice. Following your suggestions, we added two models: 1) a Ge-SO model with Ge doping but without lattice distortion (**Supplementary Figure 33b**), and 2) a lattice-distorted SO model without Ge doping (**Supplementary Figure 33c**). The results demonstrate that both the doping of Ge and the distorted SO lattice cause an upward shift in the *p*-band center of SO. However, the degree of upward shift is lower when only Ge is doped or only lattice distortion compared to the combination of both factors (**Supplementary Figure 33e-h**). This confirms that the two changes induced by the doping of Ge collectively modify the electronic structure of SO.

Revision: We have added relevant results to the manuscript. “To investigate the impact of lattice distortion and Ge electronic interaction on the p -band center of SO, we established four models: 1) a normal SO model (**Supplementary Figure 33a**), 2) a Ge-SO model with Ge doping but no lattice distortion (**Supplementary Figure 33b**), 3) a lattice-distorted SO model without Ge doping (**Supplementary Figure 33c**), and 4) a normal Ge-SO model (**Supplementary Figure 33d**). The p -band centers of these models were calculated from DOS. The results demonstrate that both the doping of Ge and the distorted SO lattice cause an upward shift in the p -band center of SO. However, the degree of upward shift is lower when only Ge is doped or only lattice distortion compared to the combination of both factors (**Supplementary Figure 33e-h**). This confirms that the two changes induced by the doping of Ge collectively modify the electronic structure of SO.”

Rev. Supplementary Figure 33. Crystal structure and electronic density of states of (a, e) SO, (b, f) Ge doped SO but without lattice distortion, (c, g) lattice-distorted SO but without Ge doping and (d, h) normal Ge doped SO.

6. The authors built a model with 25% Ge doping in SnO₂ and observed an upshift in the p band center. It would be interesting to investigate how the ratios of Ge doping affect these characteristics in SnO₂.

Response: Thank you for your valuable suggestions. Following your recommendation, we created models of SO with Ge doping levels of 0%, 6.25%, 12.5%, 18.75% and 25%, respectively. Subsequently, we investigated the p -band center of O in these models (**Supplementary Figure 32**). Our findings indicate that the p -band center of O rises proportionally with the increase in Ge doping amount, aligning with our first draft.

Revision: We have added relevant results to the manuscript. “Finally, we investigated the impact of germanium’s percentage and lattice distortion on the p band center of SO. Firstly, the impact of the germanium ratio on p -band center of SO is investigated. we investigated p -band centers of SO with Ge doping levels of 0%, 6.25%, 12.5%, 18.75% and 25%, respectively (**Supplementary Figure 32**). The findings indicate that the p -band center of O rises proportionally with the increase in Ge doping amount, aligning with our first draft.”

Rev. Supplementary Figure 32. The electronic density of states (DOS) of (a) 6.25%Ge-doped SO, (b) 12.5%Ge-doped SO and (c) 18.75%Ge-doped SO, respectively.

7. Can the lattice oxygen participate in other gas-sensing processes, such as formaldehyde, benzenes or other VOC?

Response: Thank you for your valuable suggestions. In this work, we mainly study the participation of lattice oxygen in sensing reaction. Hence, the gases that can react with chemisorbed oxygen are more reasonable candidates to reveal this mechanism. We conducted experiments to investigate responses of SGO to the gases that can react with adsorbed oxygen under both air and argon conditions, such as CO, formaldehyde (FA) and ethanol (ET). The results (**Supplementary Figure 25**) indicate that the response values of CO, FA and ET in the argon environment are improved compared to in the air environment, at operating temperature of 220°C. Therefore, we deduced that the conversion of lattice oxygen to chemisorbed oxygen may be a ubiquitous phenomenon at an operating temperature of 220°C.

Rev. Supplementary Figure 25. The response values of SGO to H₂, CO, formaldehyde, ethanol in air and argon, respectively.

8. There are some formatting errors in the text, please check it carefully.

Response: Thanks. The formatting errors have been carefully checked and improved.

Reviewer #3 (Remarks to the Author):

The authors describe sensing results to Ge-doped SnO₂ samples, attempting to rationalise the higher performance using XPS, DRIFTS and DFT calculations. However there are deficiencies in their analysis that mean I am unconvinced by some of their data, and some of their rationalisation appears to be pure speculation. They have also missed some key literature that would likely allow them to provide a more convincing discussion. Consequently I am recommending rejection of the article, and provide the detailed basis for this below (as well of notes of more general corrections), which hopefully will help the authors amend the article for future publication elsewhere.

Response: We would like to express our gratitude for your meticulous review and valuable feedback on our work. We acknowledge your concerns regarding certain data and analysis presented in our articles and appreciate your identification of potential issues and shortcomings. In response to your comments, we will address each point individually.

(1) Response to the questioning of analysis deficiencies and pure speculation.

With regards to the section where we employ XPS, DRIFTS, and DFT calculations to elucidate the high performance, we understand your apprehension that some of these explanations may be speculative. However, it is important to emphasize that these methods are widely accepted and extensively utilized in the fields of materials science and sensing to gain profound insights into material properties and sensing mechanisms. Our intention is to employ these methods to provide a plausible explanation for our experimental results and to establish the consistency of our data with existing theories. I believe we have done that. We also have included additional tests to provide further support for our conclusions. Interestingly, the response values of SGO to CO, formaldehyde, and other gases that can react with adsorbed oxygen in an argon environment are higher compared to those in air. This finding strengthens the universality of the LOM mechanism we proposed. Nevertheless, we welcome further discussion and suggestions pertaining to this aspect.

(2) Response on missed some key literature.

We apologize for the oversight of certain crucial documents. While composing the article, we made earnest efforts to encompass significant literature in the relevant field, but it is inevitable that some references may have been inadvertently omitted. We will meticulously examine the literature you have mentioned and incorporate appropriate citations and discussions in our future work, thereby enhancing the comprehensiveness and persuasiveness of our article.

(3) Our statement about article innovativeness.

Lastly, we would like to underscore the innovative nature of our investigation. Semiconducting metal oxides are commonly used as gas-sensing materials, but the lack of understanding of the dynamic evolution of the material surface during the gas-sensing reaction has hindered the development of improved gas-sensing materials through material design. Based on the above issues, we have made the following innovations: (1) we employed *in-situ* Raman and *in-situ* DRIFTS techniques to observe the dynamic evolution of the surface of gas-sensing materials during the gas-sensing reaction. Based on our observations, we propose a novel gas-sensing reaction mechanism involving lattice oxygen. (2) DFT calculations show that the rational use of this new mechanism can promote the design of higher-performance gas-sensing materials, which is where the new mechanism is superior to the traditional oxygen adsorption mechanism. (3) Importantly, this mechanism is applicable not only to our specific system but also to a wide range of semiconducting metal oxide gas-sensing materials. Our findings open up new possibilities for the rational design of

gas-sensing materials. Once again, we highly recommend our work for your consideration.

1. Line (L)40; pollution from hydrogen combustion is entirely dependent on how it was produced.

Response: Thanks for the reviewer's comment. As referee mentioned, hydrogen can be produced from many different renewable and non-renewable sources, with widely varying costs and carbon dioxide emissions. Hydrogen produced via conversion of fossil fuels (e.g. steam reforming of natural gas) can result in a large amount of carbon dioxide emissions, hence is not environmentally or climate-friendly. Low-carbon hydrogen generally includes green hydrogen (hydrogen from renewable electricity) and blue hydrogen (hydrogen from fossil fuels with CO₂ emissions reduced by the use of Carbon Capture, Utilization and Storage). Green hydrogen is carbon-free and is rapidly developing from pilot to commercial-scale operation in many parts of the world.

Revision: We have corrected the expression to "Hydrogen, a high-energy-density energy carrier which can be produced from renewable sources (solar, hydro, wind), has recently received considerable interest in various field, including new energy vehicles."

2. L42; one assumes that is the explosive concentration mixed with air?

Response: Thanks for the reviewer's comment. The explosive concentration given in the introduction is when it is mixed with air. We have also added the appropriate references [10.1016/j.pecs.2018.02.003](https://doi.org/10.1016/j.pecs.2018.02.003)

Revision: We have added this point in Line 42 as follows: the easy escape and wide range of explosive concentration (4% to 75% when mixed with air) of hydrogen highly impedes the commercialization of hydrogen vehicles.

3. L53-57; this description is largely superfluous as it is repeated later.

Revision: Thanks for the reminder! We have modified "As a result of this reaction, the electrons that were captured by the chemisorbed oxygen are released back into the conduction band (or valence band in the case of *p*-type semiconductors) of the gas-sensing material. This release of electrons causes a decrease (or increase) in the resistance of the material. With this mechanism, high sensing response can be obtained by increasing the content of chemisorbed oxygen, and this is supported by experimental findings that the sensing response of In₂O₃, SnO₂, ZnO, and so on." into "The commonly accepted mechanism for hydrogen sensing is the chemisorbed oxygen mechanism, which is largely drawn from studies on the surface of metal oxide semiconductors. In this mechanism, the hydrogen gas undergoes a redox reaction with the chemisorbed oxygen on the surface of the material, and the electrons transfer between the oxygen chemisorbed on the surface and conduction/valence band. With this mechanism, all the oxygen species participating in sensing process come from the air and high sensing response can be obtained by increasing the content of chemisorbed oxygen, and this is supported by experimental findings that the sensing response of In₂O₃, SnO₂, ZnO, and so on."

4. L78; better to refer to Ge-132 using its chemical name or formula.

Response: Thanks for the reminder! We have modified Ge-132 into Bis(carboxyethyl germanium) sesquioxide (Ge-132).

5. L83; That shift would only be relevant to the peak shown at around 26 degrees unless it is the result of a systematic error. If the unit cell has changed size then the relevant lattice parameters should be given for both rutile-SO, SGO and for a reference of SO

Response: We appreciate the invaluable suggestions. Firstly, we state that this is not a systematic error. X-ray diffraction (XRD) works on the principle that in a crystal, the atoms are periodically arranged on the crystal planes, and the phase of the scattered waves from all atoms on all crystal planes in the direction of Bragg's equation is exactly the same, and their amplitudes will be enhanced by each other, i.e., phase-length interferences will take place, which make diffraction peaks appear in the upper part, and cancel each other out in the other directions, and no diffraction peaks will appear. According to the Bragg equation ($2d\sin\theta=n\lambda$), where d is the crystal plane spacing, n is the number of reflection levels; θ is the grazing angle; and λ is the wavelength of the X-rays. When the crystal cell shrinks, the crystal plane spacing decreases, and the grazing angle will become larger in order to ensure the equations are balanced, which is consistent with our experimental results. In addition, there are also relevant references that describe this problem in detail.

Secondly, to further validate our analysis, we have refined the XRD of SO and SGO by Rietveld refinement (Supplementary Figure 1-2) and given the associated lattice parameters (Supplementary Table 1). The cell size of SO is 71.71\AA^3 and that of SGO is 70.65\AA^3 . Moreover, in the reported references, the cell size of SO is $71.4\text{-}74.21\text{\AA}^3$, which is in general agreement with our results.

Rev. Supplementary Figure 1. XRD pattern with a refinement plot of SO.

Rev. Supplementary Figure 2. XRD pattern with a refinement plot of SGO.

Rev. Supplementary Table 1. The crystal information for SO and SGO according to the XRD profile fitting results.

Chemical name	Stannic oxide	Germanium-doped Stannic oxide
Chemical formula	SnO ₂	Sn _{0.8} Ge _{0.2} O ₂
Crystal system	Tetragonal	Tetragonal
Space group	P42/mnm	P42/mnm
Cell length a (Å)	4.7417	4.7211
Cell length b (Å)	4.7417	4.7211
Cell length c (Å)	3.1894	3.1697
Cell angle α (°)	90.0	90.0
Cell angle β (°)	90.0	90.0
Cell angle γ (°)	90.0	90.0
Cell size (Å ³)	71.71	70.65

6. L86/87; the evidence for this should be shown (at least included in SI)

Response: Thanks for the comments. The crystalline size of the SGO and SO were determined by the X-ray line broadening method using the Scherrer equation $D = \frac{k\lambda}{\beta_D \cos \theta}$, where *D* is

the crystalline size in nanometers, λ is the wavelength of the radiation (1.54056 Å for CuK α radiation), *k* is a constant equal, β_D is the peak width at half-maximum intensity, and θ is the peak position. The breadth of the Bragg peak is a combination of both instrument- and sample-dependent effects. To decouple these contributions, it is necessary to collect a diffraction pattern from the line broadening of a standard material such as silicon to determine the instrumental broadening. By Scherrer equation, the crystalline size of SGO and SO are calculated to be 3.8 nm and 12.6 nm, respectively. Hence the particle size of SGO is much smaller than the nanoparticles in SO. Besides XRD, we also have compared the grain size via HRTEM in the following paragraph: “It can be observed that SGO consists of several micrometers long nanofibers assembled by nanoparticles with

particle sizes in the range of 5-7 nm, which is much smaller than the nanoparticles in SO (15 nm, **Supplementary Figure 7**).

7. L97/98; Not convinced by this - the apparently higher intensity in this region compared to SO appears more likely to be a function of the weak intensity of the A_{1g} and B_{2g} bands rather than an increase in intensity of this peak.

Response: 632 cm⁻¹ can be assigned to the A_{1g} symmetric stretching mode of Sn-O bonds. It can be viewed as a symmetrical stretching vibration of the oxygen atoms around stannum ions. The 550 cm⁻¹ band could be assigned to oxygen vacancies. The ratio of integral intensity of the band of oxygen vacancies and of the A_{2g} mode of SGO or SO [A_{550}/A_{632} , for SGO or SO samples] reflects the relative oxygen vacancy concentration. Hence, the oxygen vacancy concentration in SGO is higher than SO.

Besides Raman bands, EPR spectra also confirm this point. As shown in **Supplementary Figure 31**, the signal intensity of SGO is stronger than that of SO, indicating that the oxygen vacancy concentration in SGO is higher than SO.

Revision: We have modified this point and added following explanation in the revised manuscript: “SGO exhibits relatively stronger Raman bands at 550 cm⁻¹ than 632 cm⁻¹, indicating relative oxygen vacancy concentration (A_{550}/A_{632}) is higher than that in SO. Moreover, EPR spectra also confirm that SGO has more oxygen vacancies compared to SO (**Supplementary Figure 31**).”

8. L110-112; Need to demonstrate that this shift is not observed in, e.g. C1s or O1s peaks as well (see later point also)

Response: Thank you for your suggestions! In our study, all the binding energy in XPS peaks were referenced to the C 1s peaks at 284.8 eV of the surface adventurous carbon. Hence, the Sn 3d XPS peaks of SGO indeed exhibit a shift towards higher binding energy. We also added C 1s XPS spectrums of SGO and SO in **Supplementary Figure 3**.

Revision: We also declare in the manuscript: “To ensure the reliability of XPS data, we performed data correction using carbon peaks (**Supplementary Figure 3**).”

Rev. Supplementary Figure 3. The C 1s XPS spectrums of SGO and SO.

9. L122/123; Is this consistent with the reduced lattice size of SGO compared to SO - comparison with equivalent HRTEM of SO would be useful.

Response: We appreciate the invaluable suggestions. We have added the HRTEM image of SO in **Supplementary Figure 8**. The interplanar spacing of SO was measured to be 0.264nm, which is slightly larger than that of SGO (0.262 nm). This result is agreement with the theoretical value, in which the lattice constant for SGO is calculated to be 9.50Å*9.52Å*6.36Å and SO is 9.60Å*9.60Å*6.47Å.

Revision: We have added this point in the revised manuscript: “The high-resolution TEM (HRTEM) image (**Figure 1d**, inset, **Supplementary Figure 8**) displays a well-defined crystal structure of SGO

and SO with clear fringes, respectively. The lattice spacing of 0.262 nm and 0.264 nm corresponds to the (101) planes of SGO and SO, respectively.”

Rev. Supplementary Figure 8. HRTEM image of SO.

10. L140-142; Where is the evidence for this?

Response: We appreciate this important comment. We are sorry that our expression is not rigorous and makes our readers confused. Increasing temperatures can not cause a decrease in the number of active sites available for H₂ adsorption. For the sensing process, the operating temperature was a key parameter to gas sensing performance. When the operating temperature was changed, both the adsorption-desorption procedure and redox reaction rate would be greatly affected. With the increase in the operating temperature, the hydrogen molecules gain enough energy to overcome the activation energy barrier to react with the oxygen species. However, when the temperature is too high, gas molecules move faster and desorb before they have time to participate in the gas-sensing reaction. Hence the responses of SGO exhibit a trend of “increase-maximum-decrease” as the operating temperature increases.

Revision: We have modified the “This is due to the promotion of surface reaction rates at higher temperatures, but also a decrease in the number of active sites available for H₂ adsorption at high temperatures.” into “With the increase in the operating temperature, the hydrogen molecules gain enough energy to overcome the activation energy barrier to react with the oxygen species. However, when the temperature is too high, gas molecules move faster and desorb before they have time to participate in the gas-sensing reaction. Hence the responses of SGO exhibit a trend of “increase-maximum-decrease” as the operating temperature increases.”

11. L151/152; What is beta?

Response: Sorry for the carelessness. The term 'beta' in this context refers to the slope of the linear relationship between the logarithm of the gas concentration and the logarithm of the response value. In our study, we have adjusted the value to ($\beta_1 = 0.18$) ($\beta_2 = 0.70$), respectively.

12. L208/209; By eye the shift in binding energy of Sn 3d of around 0.4 eV in Figure 1c. If so, this change in Fermi energy would account for the change in Sn peak position, not any change in the valence state of Sn as suggested above. Comparing Figure 3e with S15, I am pretty sure there is a shift on the O1s also, which indicates a change in Fermi level and not a change in valence state of Sn.

Response: Thank you for your suggestion, it is highly valuable for us to enhance our work. The reviewers think the changes in Sn peak position originated from the changes of Fermi energy instead of valence state of Sn. However, the changes in the Fermi level and alterations in the valence state are not contradictory phenomena for the inorganic compounds. On the contrary, they are usually observed simultaneously. The electron transfer between Ge and Sn have also been confirmed by other approach: (i) the electron local function (ELF) in **Figure 5b** confirms that the electrons are transferred from Sn to Ge; (ii) Ge and Sn are located in the same group, and Ge^{4+} exhibits a higher electronegativity than Sn. It is obvious that Ge possesses stronger electron withdrawing ability than Sn (J. Phys. Chem. A, 2005, 109, 12, 2925–2936). Hence, the electron transfrom from Sn to Ge is reasonable. The reduction in electrons within the Sn^{4+} conduction band consequently results in a decrease in the Fermi level of SGO when compared to SO.

This is our interpretation of the influence of element doping on the electronic structure of materials. If you have any differing viewpoints or additional references, we welcome further discussion.

13. L217-219; This ignores the active site and the fractional occupation of that active site with oxygen. Also, it would be relatively easy to evidence this conjecture by looking at the resistance before and after H_2 exposure for SO and SGO. It is always worth including raw resistance data in SI, not just response data.

Response: Thank you for your valuable comments and suggestions, which have helped improve the clarity of our work. We have included additional resistance data of SGO and SO before and after exposure to H_2 . As shown in **Supplementary Figure 12**, the resistance of SO is higher compared to SGO in the air. This difference may be attributed to the presence of oxygen defects in SGO. The introduction of defect energy levels leads to a reduction in resistance. This has an impact on our ability to determine the quantity of adsorbed oxygen based on changes in resistance values.

To further investigate the quantity of adsorbed oxygen, we conducted XPS and O_2 -TPD experiments (**Figure 3e**, **Figure 3f**, **Supplementary Figure 23**). Our findings revealed that SGO has more surface adsorbed oxygen than SO.

Rev. Supplementary Figure 12. The resistance of SGO and SO before and after exposure to H_2 .

14. L224-226; the idea that XPS can show any information regarding oxygen vacancies, in particular for any samples that have been air exposed, has been comprehensively debunked (10.1016/j.susc.2021.121894 and 10.1021/acs.chemmater.3c00801). Consequently the entire

section on XPS needs revisiting.

Response: Many thanks for the valuable advice! XPS readily reveals the presence of oxygen in a material. Oxygen species usually generate three peaks in the XPS photoelectron binding energy spectrum. Two of them are ascribed to bulk or lattice O and oxygen bound to the surface without debate. However, the attribution of the other peak lies between above two peaks (530 eV ~ 531 eV) and is divergent. Most of the works, including highly respected materials chemists (later Nobel laureates), deem that it originated from oxygen vacancies. In few recent research (10.1016/j.susc.2021.121894 and 10.1021/acs.chemmater.3c00801), in which theoretical calculation and XPS analysis were carried out, the peak was claimed to come from hydroxyl oxygens or water molecules strongly bound to the exposed surface of metal oxides. These two works are more convictive and professional. Before writing this paper, we undoubtedly consulted a lot of related literature and ascribed it wrongly to be vacancies according to the view of the majority of literature. However, it should be noted that either hydroxyl oxygens or water molecules are chemisorbed oxygen species, so our conclusion is still reasonable. We have replaced vacancies oxygen with “hydroxyl oxygens/water molecules strongly bound to the surface of SO/SGO” and made corresponding modify in **Figure 3e** and **Supplementary Figure 23** as follows:

Rev. Figure 3e. The O 1s XPS spectrum of SGO.

Rev. Supplementary Figure 23. The O 1s XPS spectrum of SO.

15. L247/248; This seems to directly contradict the authors statements above using XPS data

Response: Thanks for your reminder! By XPS data, we concluded that the content of adsorbed oxygen species for SGO is only slightly higher than that of SO (25% vs. 16%). There is only 9% increase in the content of adsorbed oxygen due to the incorporation of Ge. However, the response

value of SGO is more than three times higher than that of SO. In line 247-248, we meant to explain that the content of adsorbed oxygen is not correlated linearly with the response value.

Revision: We have modified “the content of adsorbed oxygen is not greatly increased by the incorporation of Ge” into “the content of adsorbed oxygen is not correlated proportionably with the response value, the percentage increase in adsorbed oxygen content is much lower than the increase in response value.”

16. L293-297; the mechanism proposed is very strange. It seems to be a hybrid between a traditional gas sensing mechanism using ionosorbed oxygen and the Mars van Krevelen mechanism of oxidation used in catalysis. I am not sure why the authors want there to be charged chemical species on the surface, when their evidence is all about the oxygen vacancy concentration (e.g. EPR) – I am not expert in DRIFTS but even a quick search revealed the peak around 1050 is associated with Sn-O and therefore I see no need to invoke species they have not measured. There are several examples in literature of sensing mechanisms that are discussed only in terms of lattice oxygen/oxygen vacancies, e.g. 10.1016/0040-6090(88)90424-5, 10.1016/0167-2584(87)90918-2, 10.1021/acs.jpcc.8b01446, 10.1021/acssensors.1c01727

Response: We appreciate the valuable suggestions! Firstly, we would like to acknowledge that the mechanism proposed in our study serves as a supplement and improvement to the traditional gas sensing mechanism. In our work, we found that SGO shows a response value that remains the same or even higher under vacuum conditions compared to that in air. This phenomenon is incompatible with traditional gas sensing mechanism (adsorbed oxygen mechanism), in which adsorbed oxygen is the only species that participates in sensing process. Hence, we concluded that other oxygen species also play an important role. We firmly believe that the LOM mechanism offers several advantages over the conventional method, particularly in explaining unconventional phenomena such as the enhanced response of SGO to hydrogen under argon conditions. Moreover, based on our latest understanding, we are proud to present the first-ever proposal for gas-sensing material design through *p*-band center regulation. This milestone achievement holds immense significance in advancing the field of hydrogen-sensing materials and deepening our understanding of gas-sensing mechanisms. Besides, the sensing process has many similarities with catalysis. For example, they are both involved with adsorption and desorption processes on the surface of catalysts/oxides. It is also reasonable that a step occurred in the Mars van Krevelen mechanism can also be applied in sensing process.

Secondly, at higher operating temperatures, the presence of ionic chemically adsorbed oxygen (charged chemical species) on the SnO₂ surface has been identified through various techniques such as IR (10.1016/0584-8539(94)01216-4), TPD (10.1016/j.snb.2022.131754), EPR (10.1021/acscatal.9b02408). This observation has been confirmed and is widely acknowledged by the scientific community, including in some of the papers you provided (see the fifth line of the abstract in 10.1021/acs.jpcc.8b01446 for more details).

Thirdly, there have been reports on the measurement of chemically adsorbed oxygen ions using in-situ infrared spectroscopy (10.1016/0584-8539(94)01216-4, 10.1007/s10854-022-08004-3). The peak observed around 1050 cm⁻¹ is specifically identified as ionic chemically adsorbed oxygen (Sn-O).

Thanks again for the advice!

17. L304-306; most recent calculations on SnO₂ suggest that bulk oxygen vacancies are too deep to be significantly ionised, even at sensor operation temperatures - 10.1103/PhysRevMaterials.2.054604

Response: Thanks for the comments. Bulk oxygen vacancies are indeed too deep to be significantly ionized. The sensing process takes place on the surface of the oxides, hence we mainly discuss the relationship between sensing properties and surface features of SO and SGO instead of bulk. To clear up confusion, we have modified the term 'oxygen vacancies' into 'surface oxygen vacancies' at relevant positions in the paper.

18. L309-312; this requires far more discussion.

Response: Thanks for the suggestion. To enhance the rigor of our discussion, we have integrated EPR spectra, Raman spectrum and DFT calculations, providing a more comprehensive analysis.

Revision: We have added this discussion in the revised manuscript: “SGO exhibits relatively stronger Raman bands at 550 cm⁻¹ than 632 cm⁻¹, indicating relative oxygen vacancy concentration (A_{550}/A_{632}) is higher than that in SO. Moreover, EPR spectra also confirm that SGO has more oxygen vacancies compared to SO (**Supplementary Figure 31**). All these phenomena should originate from the electronic interaction between Ge⁴⁺ and SO. DFT calculations also demonstrate that the introduction of Ge induces distortion in the SGO lattice, thereby increasing the likelihood of defect formation.”

REVIEWER COMMENTS

Reviewer #1 (Remarks to the Author):

The revision is well addressed. I recommend the publication of revised manuscript.

Reviewer #2 (Remarks to the Author):

I believe this manuscript has been properly revised and it is satisfied with all the requested adjustments. Therefore, I can now recommend this work for publication.

Reviewer #3 (Remarks to the Author):

Except where noted below, I am happy with the response of the authors. The following would need to be addressed before publication.

Point 5; I think the authors have missed the point I was making. They say there is a shift of 0.27 degrees as a result of lattice contraction. If it is the result of lattice contraction then the size of the shift will change with peak angle, i.e. a shift of 0.27 degrees is only relevant to the single peak to which it applies. If there is a peak shift of 0.27 degrees for EVERY peak in the pattern then this suggests there is a systematic error with the measurement, e.g. an issue with height alignment. The authors need to clarify/be more precise in their description.

Point 12; the authors have ignored the comment regarding the apparent shift in binding energy of the O1s peaks between Figure 3e (B.E. O(L) \sim 530.6 eV) and Figure S23 (B.E. O(L) \sim 530.2 eV). This shift is around 0.4 eV and in the same direction as the shift they assign to electron transfer from Sn. I cannot think of an explanation for both Sn and O peaks to shift in the same direction and by the same magnitude (see for instance P17, Line 380 where their description Sn \diamond O \diamond Ge would make this impossible), other than an effect such as an increase in surface charge as a result of ionisation of surface oxygen vacancies. However this then is not consistent with the newly presented C1s spectra in Figure S3. The authors must ascribe the change in B.E. of the O1s peak to provide a consistent description of the XPS data.

Point 14; the authors are correct that the signal they originally ascribed to Ov is consistently misassigned in many contemporary papers – it is one of the major issues affecting XPS analysis currently. I note that they are still assigned as oxygen vacancies on P11 L251, which should be amended.

Point 16; just a comment – I think it strange that the authors understand that the formation of surface oxygen vacancies will lead to the change in measured resistance observed but still want there also to be surface oxygen species. They have chosen to be consistent with prevailing literature in the gas sensing field (ionosorption) rather than consistent with the catalysis literature (MvK). It's their choice, but it seems like a missed opportunity to me to bring together descriptions of gas sensing with descriptions of catalysis.

Additional: In looking in detail at Figure S33 it appears that the Sn 5p states also drift closer to the Fermi level (also apparent in Figure 5d and e). The authors argument is that the O 2p states shift closer to the Fermi level, but if the Sn 5p states do also move, would it not be better to depict this as the Fermi level moving down rather than the states all moving up? Also in being explicit that the PDOS plots are both plotted relative to the independent Fermi levels and hence it cannot be inferred that the O 2p states in SGO are absolutely higher than the 2p states in SO, despite the graphic in 5f inferring this.

RESPONSE TO REVIEWERS' COMMENTS

Reviewer #1 (Remarks to the Author):

The revision is well addressed. I recommend the publication of revised manuscript.

Response: We are grateful for the reviewer's positive recommendation in acceptance of our manuscript and insightful comments that help greatly improve the quality of the manuscript.

Reviewer #2 (Remarks to the Author)

I believe this manuscript has been properly revised and it is satisfied with all the requested adjustments. Therefore, I can now recommend this work for publication.

Response: We are grateful for the reviewer's positive recommendation in acceptance of our manuscript and insightful comments that help greatly improve the quality of the manuscript.

Reviewer #3 (Remarks to the Author):

Except where noted below, I am happy with the response of the authors. The following would need to be addressed before publication.

Response: We thank the reviewer's additional comments here! We have addressed the comments point-by-point as follows.

1. Point 5; I think the authors have missed the point I was making. They say there is a shift of 0.27 degrees as a result of lattice contraction. If it is the result of lattice contraction then the size of the shift will change with peak angle, i.e. a shift of 0.27 degrees is only relevant to the single peak to which it applies. If there is a peak shift of 0.27 degrees for EVERY peak in the pattern then this suggests there is a systematic error with the measurement, e.g. an issue with height alignment. The authors need to clarify/be more precise in their description.

Response: We apologize for the misunderstanding in our initial response. We have provided a comparison of the XRD diffraction peak positions and shift between SO and SGO in **Supplementary Table 2**. The peak offset of SGO is increasing as the diffraction angle increases, which aligns with Bragg's equation. This demonstrates that the XRD diffraction peak shift in SGO is attributed to lattice shrinkage rather than a systematic error.

Rev. Supplementary Table 2. The peak shift information for SO and SGO according to the XRD patterns. (# means that the diffraction peaks at this position are too broadened to be precisely localized)

Number	2 Theta-SO (°)	2 Theta-SGO (°)	Peak shift (°)
1	26.45	26.72	0.27
2	33.75	34.04	0.29
3	37.82	38.18	0.35

4	38.85	#	-
5	51.70	52.18	0.48
6	54.63	#	-
7	57.79	#	-
8	61.86	62.39	0.53
9	64.66	#	-
10	65.86	#	-

2. Point 12; the authors have ignored the comment regarding the apparent shift in binding energy of the O1s peaks between Figure 3e (B.E. O(L) ~ 530.6 eV) and Figure S23 (B.E. O(L) ~ 530.2 eV). This shift is around 0.4 eV and in the same direction as the shift they assign to electron transfer from Sn. I cannot think of an explanation for both Sn and O peaks to shift in the same direction and by the same magnitude (see for instance P17, Line 380 where their description Sn O Ge would make this impossible), other than an effect such as an increase in surface charge as a result of ionisation of surface oxygen vacancies. However this then is not consistent with the newly presented C1s spectra in Figure S3. The authors must ascribe the change in B.E. of the O1s peak to provide a consistent description of the XPS data.

Response: Many thanks for pointing out the inaccurate explanation.

We acknowledge the shift of O1s peaks still exists even corrected by C 1s peaks, we concur with your viewpoint that the shift is a result of the reduction of the Fermi level. However, the correction through the C 1s peak is valid for two reasons. Firstly, the C 1s signal typically originates from conductive adhesive, ensuring stability and reliability for all non-carbon-based materials. Secondly, the calibration based on the C 1s peak is widely accepted in the field.

Based on your suggestion, we calibrated the Sn 3d peak using the position of lattice oxygen and still observed a slight shift in the Sn 3d peak of SGO (**Rev. Figure**). Therefore, it is likely that the change in Sn valence state is a secondary factor contributing to the observed shift in the Sn 3d peak.

To sum up, we have modified our description accordingly as follows: “Compared with SO, the Sn 3d peaks of SGO exhibit a slight shift toward higher binding energy, we hypothesize that it can be attributed to a synergistic effect arising from the downward shift of Fermi level and an increase in valence of Sn species in SGO.”

Rev. Figure. Sn 3d XPS spectra of SGO and SO, calibrated by O 1s.

3. Point 14; the authors are correct that the signal they originally ascribed to Ov is consistently misassigned in many contemporary papers – it is one of the major issues affecting XPS analysis currently. I note that they are still assigned as oxygen vacancies on P11 L251, which should be amended.

Response: Sorry for the carelessness. We have modified the oxygen vacancy to hydroxyl oxygen.

4. Point 16; just a comment – I think it strange that the authors understand that the formation of surface oxygen vacancies will lead to the change in measured resistance observed but still want there also to be surface oxygen species. They have chosen to be consistent with prevailing literature in the gas sensing field (ionosorption) rather than consistent with the catalysis literature (MvK). It's their choice, but it seems like a missed opportunity to me to bring together descriptions of gas sensing with descriptions of catalysis.

Response: We appreciate your sincere suggestions and acknowledge that there may be differences between our understanding of the gas sensing mechanism. Your understanding of the gas-sensing mechanism from the perspective of thermocatalysis is highly insightful and demonstrates creativity. We have also incorporated several ideas from MvK, as you mentioned, to enhance our research. At present, your differences with us could be attributed to the limitations of current testing technology.

We believe that as testing technology and theoretical advancements continue to progress, this mystery can be eventually resolved, and we will have shared understanding in the future.

However, we currently maintain that our analysis is reasonable, substantiated by our experimental results. Our infrared spectroscopy and XPS spectrum have objectively observed the presence of surface adsorbed oxygen, which is widely acknowledged in the gas sensitivity field.

5. Additional: In looking in detail at Figure S33 it appears that the Sn 5p states also drift closer to the Fermi level (also apparent in Figure 5d and e). The authors argument is that the O 2p states shift closer to the Fermi level, but if the Sn 5p states do also move, would it not be better to depict this as the Fermi level moving down rather than the states all moving up? Also in being explicit that the PDOS plots are both plotted relative to the independent Fermi levels and hence it cannot be inferred that the O 2p states in SGO are absolutely higher than the 2p states in SO, despite the graphic in 5f inferring this.

Response: Thank you for your suggestion, it greatly improves our work! We agree with your point. Combining our experimental results of Work Function and the DOS plot, the Fermi level of SGO indeed moves down, compared with SO. However, it is the difference between the O 2p-band center and the Fermi level that decides the activity of surface oxygen exchange kinetics (neither the Fermi energy level nor O 2p alone). As Fermi level is a well-known physical concept, the Fermi level is always set to 0, to give a direct and convenient sight for the comparison of the O 2p-band center between different materials. (Nat. Commun., 4.1 (2013): 2439; Nat. Commun., 2021, 12(1): 3992.). And in our discussion, SGO has a higher O 2p-band center (relative to Fermi level), than SO, hence the surface lattice oxygen of SGO is more favorable to transform to chemisorbed oxygen.

In response to your suggestions, we have modified our description accordingly as follows: “Moreover, SGO has a higher O 2p-band center (relative to fermi level) than SO (-2.23 eV vs -3.07 eV), suggesting that the transformation from lattice oxygen to chemisorbed oxygen are thermodynamically more favorable of SGO (**Figure 5f**). This further implies that introducing Ge and the resulting lattice distortion contribute to the optimizing of the electronic structure, as discussed in detail in the Supplementary Information (**Supplementary Figure 32-33**).”

REVIEWERS' COMMENTS

Reviewer #3 (Remarks to the Author):

I am happy with the authors latest revisions and happy for the editor to decide on publication

RESPONSE TO REVIEWERS' COMMENTS

Reviewer #3 (Remarks to the Author):

I am happy with the authors latest revisions and happy for the editor to decide on publication.

Response: We are grateful for the reviewer's positive recommendation in acceptance of our manuscript and insightful comments that help greatly improve the quality of the manuscript.